# Correlation between acoustic divergence and phylogenetic distance in soniferous European gobiids (Gobiidae; *Gobius* lineage)

Sven Horvatić[1], Stefano Malavasi[2], Jasna Vukić[3], Radek Šanda[4], Zoran Marčić[1], Marko Ćaleta[5], Massimo Lorenzoni[6], Perica Mustafić[1], Ivana Buj[1], Lucija Onorato[1], Lucija Ivić[1], Francesco Cavraro[2], Davor Zanella[1] *

**1** Department of Zoology, Faculty of Science, University of Zagreb, Zagreb, Croatia, **2** Department Environmental Sciences, Informatics and Statistics, Cà Foscari, University of Venice, Venezia Mestre, Italy, **3** Department of Ecology, Charles University, Prague, Czech Republic, **4** Department of Zoology, National Museum, Prague, Czech Republic, **5** Faculty of Teacher Education, University of Zagreb, Zagreb, Croatia, **6** Department of Chemistry, Biology and Biotechnologies, University of Perugia, Perugia, Italy

* davor.zanella@biol.pmf.hr

**Data Availability Statement:** All relevant data are within the paper and its S1 and S2 Figs, S1 Data, S1–S4 Tables files.

## Abstract

In fish, species identity can be encoded by sounds, which have been thoroughly investigated in European gobiids (Gobiidae, *Gobius* lineage). Recent evolutionary studies suggest that deterministic and/or stochastic forces could generate acoustic differences among related animal species, though this has not been investigated in any teleost group to date. In the present comparative study, we analysed the sounds from nine soniferous gobiids and quantitatively assessed their acoustic variability. Our interspecific acoustic study, incorporating for the first time the representative acoustic signals from the majority of soniferous gobiids, suggested that their sounds are truly species-specific (92% of sounds correctly classified into exact species) and each taxon possesses a unique set of spectro-temporal variables. In addition, we reconstructed phylogenetic relationships from a concatenated molecular dataset consisting of multiple molecular markers to track the evolution of acoustic signals in soniferous gobiids. The results of this study indicated that the genus *Padogobius* is polyphyletic, since *P. nigricans* was nested within the Ponto-Caspian clade, while the congeneric *P. bonelli* turned out to be a sister taxon to the remaining investigated soniferous species. Lastly, by extracting the acoustic and genetic distance matrices, sound variability and genetic distance were correlated for the first time to assess whether sound evolution follows a similar phylogenetic pattern. The positive correlation between the sound variability and genetic distance obtained here emphasizes that certain acoustic features from representative sounds could carry the phylogenetic signal in soniferous gobiids. Our study was the first attempt to evaluate the mutual relationship between acoustic variation and genetic divergence in any teleost fish.

**Funding:** The author(s) received no specific funding for this work.

**Competing interests:** The authors have declared that no competing interests exist.

## Introduction

Many animal species use sounds, together with other communication signals, to express their behaviour, and by actively changing their acoustic properties, they can control the information content of these signals [1, 2]. Interspecifically, sounds can encode the identity of the signalling individual [3–5] and for related species in sympatry, this is an important discrimination trait during reproductive interactions [6–8]. Given the significant role of sound production in the species recognition process, it is believed that divergence in acoustic signals could drive speciation [9–12].

One of the central questions in bioacoustics, when it comes to divergence caused by variations in acoustic signals, is to determine which evolutionary forces have generated prominent interspecific differences among animal taxa. Even though a signal evolution is rarely explained by a single evolutionary force, most studies invoke two common forces generally thought to be responsible for acoustic divergence: deterministic and stochastic. Deterministic (or "adaptive") forces, such as habitat adaptation [13, 14], divergence in morphology [15, 16] or selection for species recognition ("reproductive character displacement" or sometimes even sexual selection) [17–19] act to amplify the signal variations already present among species. These forces generate straightforward predictions about the direction of evolution [20]. In these circumstances, an absence of association between genotype and acoustics highlights the importance of deterministic factors and other selective pressures in shaping acoustic traits. On the other hand, stochastic (or "neutral") forces, such as sexual or social selection [21, 22] or more commonly, genetic drift and mutation [13, 23–25] could be the driving initiators for signal divergence. These mechanisms make signal divergence a highly stochastic and unpredictable process [20], where a positive correlation is usually observed between sound divergence and genotype. Likewise, some studies indicate that the two forces sometimes interact and act mutually, causing overall signal divergence [for references, see 20]. Accordingly, studies on animal sounds support both the association between acoustic variation and genetic divergence [20, 25–30] and the lack of association between the two divergences [13, 31, 32].

The acoustic repertoire among teleosts has been thoroughly investigated. In the gobioids (Gobiiformes; Gobioidei), soniferous species produce different types of acoustic signals, presenting an extensive repertoire. This acoustic repertoire shows great variability at both the inter- and intraspecific levels, with four different sound types [thump, pulsatile (drum), tonal, and complex] recorded to date, emitted mainly but not exclusively, by males as part of the breeding and aggressive behavioural displays (e.g., in some species, females are also able to produce sounds during aggressive intrasexual interactions [33]). Specifically, since male sounds are produced during reproduction for inter- and intrasexual interactions, as in painted goby (*Pomatoschistus pictus* (Malm, 1865)) and monkey goby (*Neogobius fluviatilis* (Pallas, 1814)) [33–35], they are important for evolutionary studies examining the processes leading to species radiation. Due to the morphological similarities and the lack of morphological sonic specialisations (such as ridged pectoral spines, pharyngeal teeth, enhanced pectoral fin tendons or sonic muscles attached to the swim bladder) in the investigated species [36, 37], it can be expected that gobies utilise similar acoustic components for sound production, which could reveal a certain phylogenetic pattern.

Taxonomically, the Gobioidei is one of the largest vertebrate suborders, including several families [38–40]. The European gobies belong to two of these families, Gobiidae and Gobionellidae [41]. Three lineages of European gobies have been recognized by previous phylogenetic analyses, with the *Gobius-* and *Aphia*-lineage as part of the family Gobiidae (gobiine-like clade in [42]), while the *Pomatoschistus*-lineage was nested within the family Gobionellidae (gobionelline-like clade in [42]). Traditionally, molecular studies have strongly emphasized that the

European gobies from the *Gobius*-lineage form a monophyletic group [40, 42, 43] and within this lineage, the genus *Gobius* (including *Zosterisessor*; [40, 42, 44, 45]) and endemic goby species from the Ponto-Caspian region (e.g., genera *Babka*, *Benthophilus*, *Mesogobius*, *Neogobius Proterorhinus*, *Ponticola*) [40, 42, 46] are by far the most speciose groups. To date, sound production in gobioids was most commonly documented within the *Gobius* lineage, with 11 species proving to be soniferous during behavioural trials [8]. In the *Gobius* lineage, interspecific sound diversification is thought to be highly important, since closely related species do not overlap in their acoustic features and they have the ability to produce different kinds of sounds. It was proposed that those differences follow the phylogenetic pattern to a certain degree [34, 47–50]; however, these studies were empirically limited, since no one has compared acoustic signals with genetic data to corroborate possible concordance. Furthermore, these studies have supported findings from other fish groups, such as, Malawi cichlids (Cichlidae), toadfishes (Batrachoididae), *Dascyllus* damselfish (Pomacentridae) or piranhas (Serrasalmidae) [51–54], showing that some acoustic features could be a reliable species or individual identifier. Likewise, recently documented sound emission in Amur sleeper *Perccottus glenii* (Odontobutidae) suggests a deeper sound production ancestry within the gobioids [55]. Since gobies are widely distributed in European waters [56–58] and many species live in sympatry with at least one other species, communication signals (including sounds) likely play a significant role in mating recognition and prevention of hybridisation. Therefore, the observed acoustic diversity and sound utilisation during reproduction indicate that acoustic signals could have a prominent role in the evolution and speciation of the European gobiids. There are not existing comparative studies to incorporate a correlation between acoustic signals and molecular markers (DNA fragments/gene sequences) between closely related species in Gobiidae. Therefore, the object of this study was to give a quantitative evaluation of the relationship between interspecific acoustic variation and genetic divergence in soniferous *Gobius* lineage species and, according to the observed association, discuss the potential evolutionary forces promoting sound divergence.

By combining quantitative bioacoustics and multiple molecular markers, this study examines previously documented but never comprehensively analysed acoustic diversity of the representative sounds produced by nine gobiid species (Gobiidae, *Gobius* lineage) and explores the degree to which the affinities in acoustic signals between closely related species could be related to their phylogenetic relationships. Specifically, this study aimed to: *i.* explore the interspecific acoustic variation among nine *Gobius* lineage gobiids and assess the acoustic variables responsible for the species differentiation; *ii.* investigate the phylogenetic relationship between soniferous *Gobius* lineage gobiids; *iii.* examine the correlation between interspecific acoustic divergence and genetic distance based on multilocus data (two mitochondrial and two nuclear genes) to explore the phylogenetic significance of acoustic signals in species diversity; *iv.* according to the observed correlations, discuss the evolutionary forces driving acoustic divergence in these taxa, and *v.* construct phylogenetic hypotheses on the evolution of acoustic signals in soniferous gobioids.

## Material and methods

### Study species

This study analysed acoustic signals and our species composition was based on the availability of previously recorded audio tracks enabling the comparison of nine soniferous gobiids (Gobiidae, *Gobius* lineage) belonging to five genera (*Gobius*, *Padogobius*, *Zosterisessor*, *Neogobius* & *Ponticola*; Table 1). Amur sleeper *Perccottus glenii* is a soniferous Eurasian gobioid belonging to the family Odontobutidae. Due to the sister phylogenetic position of sleepers to

**Table 1. Species of gobiids used in the acoustic-genetic analyses (together with outgroup *P. glenii*) and the collection site from which the individuals were used for acoustic analyses.** GenBank® accession numbers of sequences correspond to the genetic markers for each species used in phylogenetic analysis. Accession numbers of the sequences produced in this study are indicated.

| Species | Collection site of soniferous species | GenBank accession no. | | | |
|---|---|---|---|---|---|
| | | *cytb* | *cox1* | *rag1* | *rho* |
| *Perccottus glenii* | Danube River, Veliko Gradište, SRB | this study* | AY722171 | KF415837 | KX224234 |
| *Gobius cobitis* | Venice Lagoon, IT | this study** | KR914767 | this study** | this study** |
| *Gobius paganellus* | Venice Lagoon, IT | this study** | KR914777 | this study** | this study** |
| *Gobius niger* | Venice Lagoon, IT | KF415583 | KR914775 | FJ526891 | MW195522 |
| *Zosterissesor ophiocephalus* | Venice Lagoon, IT | EU444670 | MT670254 | FJ526851 | this study** |
| *Padogobius nigricans* | Arbiola Stream, Lucca, IT | this study** | KJ554001 | this study** | this study** |
| | Serchio River, Lucca, IT | | | | |
| *Padogobius bonelii* | Stirone Stream, Parma, IT | this study** | KJ554527 | this study** | MW195526 |
| *Neogobius fluviatilis* | Kupa-Kupa Channel, Donja Kupčina, CRO | this study* | FJ526807 | EU444718 | MW195524 |
| *Neogobius melanostomus* | Sava River, Rugvica, CRO | this study* | FJ526801 | FJ526857 | JF261593 |
| *Ponticola kessleri* | Drava River, Osijek, CRO | FJ526770 | FJ526825 | FJ526879 | MW195527 |

Abbreviations: SRB, Serbia, CRO, Croatia, IT, Italy.

A single asterisk (*) indicates the genetic samples (i.e., sequences) that come from the exact location as soniferous individuals, while two asterisks (**) indicate sequences originating from a different geographic location compared to the acoustic material.

the rest of the gobies including Gobiidae [41, 42], *P. glenii* served as an outgroup in the analyses. Traditionally, these nine gobiids have been split into two groups, Atlantic-Mediterranean (AM) and Ponto-Caspian (PC) [58]. In the study, the AM gobies include the genera *Gobius*, *Zosterisessor* and *Padogobius*. Among them, black goby *Gobius niger* Linnaeus, 1758, giant goby *G. cobitis* Pallas, 1814, rock goby *G. paganellus* Linnaeus, 1758 and grass goby *Zosterisessor ophiocephalus* (Pallas, 1814) are marine/brackish inhabitants. *Gobius niger* usually occupies similar muddy habitats as *Z. ophiocephalus*, while the two *Gobius* species (*G. paganellus* and *G. cobitis*) appear sympatric in rocky habitats (pers. obs.). Two *Padogobius* species (Padanian goby *Padogobius bonelli* (Bonaparte, 1846) and Arno goby *P. nigricans* (Canestrini, 1867) are purely freshwater species, from the Tyrrhenian (*P. nigricans*), and northern Adriatic drainages (*P. bonelli*). *Gobius paganellus*, *G. cobitis* and two *Padogobius* species usually occupy stony/pebble substrates, while *Gobius niger* and *Zosterisessor ophiocephalus* can be found on sandy/muddy bottoms [58, 59]. The PC species are mostly brackish to freshwater residents; bighead goby *Ponticola kessleri* (Günther, 1861) occupies stony or gravel habitat, similarly to monkey goby *Neogobius fluviatilis* (Pallas, 1814) which is common on gravel or sandy substrates. Round goby *Neogobius melanostomus* (Pallas, 1814) is common on a wide range of substrates [56, 57, 60]. In several Croatian watercourses, PC gobies live sympatrically and occupy similar bottom types (pers. obs.). The data regarding morphological traits (number of vertebrae and swim bladder presence) were obtained from the available literature [56, 57, 61] or from field observations (habitat). For the total number of recorded individuals per species (N) and number of analysed sounds (n), see S1 Table.

## Genomic sampling and phylogenetic analyses

DNA was extracted from fin clips preserved in 96% ethanol using a Geneaid® DNA Isolation Kit. For some species from the present study, genetic samples (*cytb* sequences) do not correspond directly to the soniferous individual (i.e., they do not belong to the fish used for sound analysis), while for some species this was the case (Table 1). For other genes (*cox1*, *rag1* and *rho*), sequences were designed by the authors or were taken from GenBank®, and they do not

correspond directly to the soniferous individual. Samples were amplified for mitochondrial genes cytochrome b (*cytb*) and cytochrome c oxidase subunit I (*cox1*), and for nuclear genes Recombination activating gene 1 (*rag1*) and Rhodopsin (*rho*). These four genes were chosen here due to their wide application in phylogenetic goby studies [42, 45, 46, 62]. *Cytb* and *rag1* were amplified according to the protocol described in [63] using either primers AJG and H5 [64] or GluF and ThrR [65] for *cytb* and RAG1F1 and RAG1R1 [66] for *rag1*. *Cox1* was amplified with primers FishF1 and FishR1 [67] according to the PCR protocol of [68]. *Rho* was amplified with primers RhodF and RhodR [69]. The PCR protocol followed [62]. PCR products were purified with ExoSAP-IT and sequencing was performed by Macrogen Europe (Netherlands) using amplification primers. The remaining sequences were downloaded from GenBank® [42, 43, 45, 46, 62, 68, 70–72] (Table 1). Sequences were visually checked in Chromas v2.6.4 and aligned in Bioedit v7.2.6.1 [73]. New sequences were deposited in GenBank® (Table 1). The phylogenetic reconstruction analyses were conducted on a concatenated dataset of all genes. Concatenation was recently evaluated as an appropriate method [74, 75]. Prior to analysing the sequence data, the best-fit model of nucleotide substitution for each molecular marker and subset of positions inside the codons was determined by PartitionFinder 2 [76–78], according to Bayesian information criterion (BIC) and under the all models option. The assessed partitioning scheme and evolutionary models are listed in S2 Table. Bayesian Inference (BI) and Maximum Likelihood (ML) approaches were used to estimate the phylogenetic relationships between the species. BI was conducted in MrBayes v3.2.2 [79] with four independent MCMC runs for 2 million generations, applying the partitioning scheme and model settings assessed by PartitionFinder 2. Trees were sampled every 1000 generations. The convergence of runs was analysed and visualised in TRACER v1.7.0. The first 25% of sampled trees were discarded as burn-in. The remaining trees were used to construct a 50% majority-rule consensus tree. Randomized Axelerated Maximum Likelihood [RAxML 8.2.12, 80] was used to assess ML, using Science Gateway portal CIPRES [81]. Partitioning scheme assessed by PartitionFinder 2 was applied. Support of nodes was estimated by applying 1000 nonparametric bootstrap replicates. Genetic distances (uncorrected p-distances) were assessed with MEGA 6 [82]. Nucleotide composition homogeneity within genes was tested with PAUP* 4.0b10 [83]. To study the history and evolution of acoustic signals in soniferous *Gobius* lineage gobiids, we coded the different sound types produced by the species from the present study into categorical characters (Character 1—"Sound type") with character states ranging from 0–3 (0—thump; 1—pulsatile; 2—tonal; 3—complex). Character states were obtained from the literature [34, 48–50]. By utilising the BI phylogenetic tree inferred from our concatenated dataset, and by including the character states for terminal taxa (outgroup—*Perccottus glenii*; ingroup—nine *Gobius* lineage gobies), we used Mesquite (v3.61) to reconstruct the character states at ancestral nodes of the cladogram. Since polymorphic states were present for the categorical character "Sound type" (i.e., some species produced several different sound types), we used the Parsimony method combined with the unordered model of evolution for the ancestral state reconstruction, under Trace Character History method in Mesquite.

## Sound recordings and bioacoustic analyses

All sounds were previously recorded and described by the authors but were not previously assembled into a single comprehensive phylogenetic framework. For all investigated taxa, audio recordings were obtained from laboratory studies [33, 47, 48, 55, 84–86], and the recording protocols and acoustic terminology were adopted as closely as possible to allow for interspecific comparison. Our acoustic dataset consists of 67 soniferous *Gobius* lineage individuals (min–max: 4–15 individuals; for nine species, see S1 Table) for which at least ten sounds were

recorded per individual and the individual means for each variable were calculated (mean ± SD = 87.0 ± 33.7 sounds analysed per species). Briefly, sounds were recorded exclusively from males under laboratory conditions during the reproductive season (all gobies spawn from early spring to late summer), using different audio equipment consisting of a hydrophone (Gulton Industries, HTI 94 SSQ or H2A-XLR) with preamplifiers (B&K 2626 or IRIG PRE) connected to a portable audio recorder (ZOOM H4n, Sony D7 or Tascam Linear PCM). Sounds were monitored and recorded during the "intruder test", where one individual exhibiting highly territorial behaviour after one week of acclimatization to laboratory conditions was marked as the resident fish occupying the shelter, in order to elicit inter- (male-female) or intrasexual (male-male) interactions. The soniferous individual was recognized during the experiments according to the presented behaviour, colour pattern or other body characteristics (fin shape or mouth colour). After recording, sounds were digitized (.wav format) and analysed using AVISOFT SASLab Pro Software (v5.2.14., Berlin, Germany) which allowed for calculation of the acoustic variables important for further interspecific acoustic comparison. In addition, the spectrogram, oscillogram and power spectrum were prepared using AVISOFT (Hamming window, 512-points FFT, resolution 7 Hz). Only sounds with a good signal-to-noise ratio were used in the analysis. Most investigated species produce only one type of sound, while some gobies (*P. glenii* and *G. paganellus*) produce two sounds or even possess an elaborate acoustic repertoire composed of three different sounds (*P. bonelli*). However, the criteria used in this study for all species imply only one representative sound type per species for further comparative analysis. The representative sound type for each species was selected based on the overall number of sounds observed in the audio recordings, i.e., the sound type most frequently registered and recorded during the behavioural trials. Six acoustic properties describing the temporal and spectral structure of gobiid sounds were calculated. Temporal parameters were sound rate (SR, sounds/min), number of pulses (NP), duration (DUR, milliseconds), and pulse repetition rate (PRR, dividing number of pulses with duration, in hertz). Peak frequency (PF, highest peak in power spectrum; hertz) and frequency modulation (FM) calculated as the difference between final PRR and initial PRR and expressed in hertz, were spectral variables in our analyses. The main purpose of the acoustic analysis, based on the representative sound types, was to construct an acoustic dataset ready for pairwise comparison with genetic divergence.

## Comparison between acoustic and genetic data

To assess whether the acoustic interspecific differences in gobiids were related to phylogenetic relationships, we investigated the association between sound divergence and genetic distance using the Mantel test as a prior choice [87, 88]. For the correlation, we used acoustic distance matrix constructed from Cluster analysis (Joining tree analysis) in STATISTICA® (v13.6.0., TIBCO, USA), in which clustering was performed with nine Gobiidae species as a grouping variable and six acoustic features as the analysis variables (dimensions). For the amalgamation (aggregation) rule, we used unweighted pair-group average (UPGMA) linkage, while the distance matrix was computed from the means of all sound variables for each species and built using the City-block (Manhattan) distance metric procedure. Genetic distance matrix was assessed using the uncorrected p-distance method in MEGA (version 10.0.5., USA), based on the concatenated dataset for all used molecular markers (i.e., mitochondrial and nuclear genes), and separately for mtDNA (*cytb* and *cox1*) and nDNA markers (*rag1* and *rho*). We used the bootstrap variance estimation method with 1,000 replications and p-distance as a substitution model for constructing the pairwise distance between the taxa. The Mantel test was conducted in *PASSaGE* v2 [89] on 9x9 distance matrices with

10,000 permutations. Likewise, two additional Mantel tests were performed between the obtained acoustic distance matrix and genetic distances based on 1) mtDNA (*cytb* and *cox1*) and 2) nDNA (*rag1* and *rho*) markers. These correlations were performed to investigate the relationship between sound divergence and genetic distance using molecular markers with different rates of mutation or evolution (mtDNA experiences a higher mutation rate than nDNA) and therefore, they could reveal different aspects of the speciation history of the examined taxa.

## Statistical analyses

Descriptive statistics were calculated for each temporal (SR, NP, DUR, PRR) and spectral (PF and FM) property of the acoustic signal produced by each species. For the preliminary explorations, we considered all sound variables except in the case of *P. glenii*, where it was not possible to calculate the FM of the thump sounds (as those sounds were not modulated in frequency), so this parameter was excluded from the comparative analyses. We transformed the overall acoustic dataset and tested it for the distribution fitting. First, continuous variables were $\log_{10}$-transformed (DUR, PRR, FM and PF), while discrete variables (SR and NP) were square root-transformed. We then tested the variables for normal distribution by using Shapiro-Wilks *W* test with a level of significance $P < 0.05$. To investigate sound variation among the *Gobius* lineage gobies, species were used as grouping variable. Since some acoustic variables are known to be influenced by emitter size, all sound features (used as means per individual) were divided by the body size (total length) following the formula ("$XTL^{-1}$", where "$X$" is the acoustic variable) proposed by [90], in order to allow appropriate interspecific comparison and to reduce the effect of fish size on acoustic variability. Since the assumption of normality was not met, for interspecific comparison we used the non-parametric Kruskal-Wallis *H* test followed by Dunn's multiple comparison test (level of significance $P < 0.001$) to investigate the variation of individual means for each sound variable across species. Individual mean values of sound variables were tested for correlation using the non-parametric *Spearman* correlation (level of significance $P < 0.001$) to investigate their mutual relationships. Furthermore, to quantify acoustic variability among the species, we applied multivariate exploratory techniques. Individual means of five sound variables of nine gobiids were compared to test for the overall signal similarity using Principal component analysis (PCA). PCA (in our case based on the correlation matrix) creates a factor space for a set of variables, and therefore we used it specifically to identify the acoustic parameters that explain the most variance among the taxa in the obtained factor space. For the interpretation of PCA results, we used as many factors as the number of eigenvalues > 1.0. In order to discriminate the species according to the acoustic parameters, a forward stepwise Discriminant function analysis (sDFA) was also carried out on the individual mean acoustic variables, with the specific aim to determine which parameters are responsible for species differentiation. In addition, sDFA was also used to assess the probability (classification rate %) at which individual sounds will be classified into the correct taxa. Specifically, sDFA enters variables into the discriminant function model one by one, always choosing the variable that makes the most significant contribution to the discrimination model. Factor structure coefficients were chosen to indicate the correlations between the variables and the discriminant functions. Partial Wilks' Lambda was chosen to indicate the contribution of each variable to the overall discrimination between species. The selection criterion for an acoustic parameter to be entered was F = 1.0, while F = 0.0 ($P = 0.01$) was the exclusion criteria for removal from the analysis. All statistical analyses were performed in STATISTICA® (v13.6.0., TIBCO, USA) software.

## Authorisations

Since all the acoustic data were already published in previous papers, no experimental acoustic work was conducted within the present study. However, all the previous experiments described in this article were compliant with the current laws for animal experimentation in Croatia (Bioethics and Animal Welfare Committee, Faculty of Science, University of Zagreb; permit #251-58-10617-18-14) and with the Venice Declaration (Italy). In addition, the licences 525-13/0545-18-2 and 525-1311855-19-2 (Ministry of Agriculture) permitted the field sampling of Croatian ichthyofauna and permits issued by Regione Veneto (Italy) for scientific fishery of Italian species. As regarding *P. nigricans*, sampling protocols were established in compliance with ethical standards, as approved by the Italian regulations and by local permitting authorities (Umbria Region), who provided the sampling authorizations (Resolution of the Regional Council (DGR) N. 19, session of 16/01/2017). All experiments were performed in accordance with standard ethological and bioacoustics procedures (avoiding suffering or damaging of fish body parts), meaning that all tested fish, after the laboratory analyses, were returned safely and unharmed to their natural habitat.

## Results

### Interspecific acoustic variation and sound properties

*Perccottus glenii* produces thumps sounds, with an irregular waveform and a lack of frequency modulation. However, the representative sound types produced by nine soniferous gobiids share certain common characteristics of their acoustic repertoire, allowing for interspecific comparison in PCA and DFA (Fig 1, S1 Table). All acoustic variables differed significantly among the species (Kruskall-Wallis test: SR: $H = 46.16$, $P < 0.001$; DUR: $H = 60.41$, $P < 0.001$; NP: $H = 64.64$, $P < 0.001$; PRR: $H = 55.22$, $P < 0.001$; PF: $H = 48.05$, $P < 0.001$; FM: $H = 60.95$, $P < 0.001$; d.f. = 9, n = 73 for each sound property), with at least one species differing from the remaining taxa according to the acoustic variables (Fig 2). The same pattern was observed even after removing the effect of fish size (TL in mm) on the acoustic variables by dividing them by TL (Kruskal-Wallis test, $P < 0.001$; d.f. = 9, n = 73 for each sound property). Correlation analysis performed on individual mean values of acoustic variables indicated that DUR and NP were mutually and significantly correlated (*Spearman* r = 0.88, n = 67, $P < 0.001$;

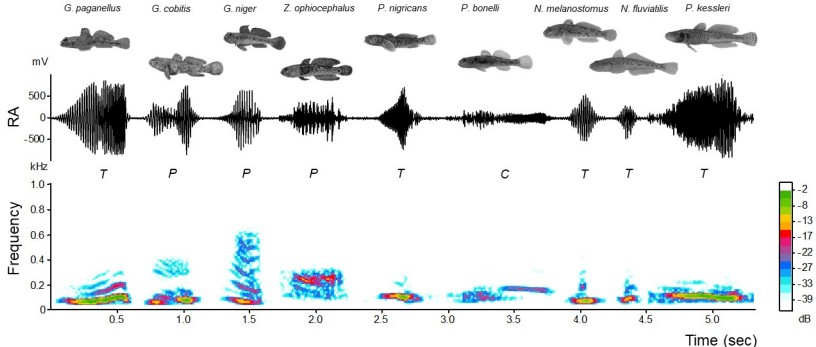

**Fig 1. Spectrograms (below: FlatTop window, 512-points FFT; 100% frame size; 93.75% overlap) and oscillograms (above) of representative sound types produced by nine soniferous gobiids (*Gobius* lineage).** Uppercase letters indicate different sounds types (*T*—tonal; *P*—pulsatile and *C*—complex). Sounds were aligned using Audacity, and the sampling frequency was converted (4000 Hz format, 16-bit accuracy) and bandpass filtered (0.05–0.5 Hz filter) in AVISOFT software. Relative intensity (dB) was included as a colour scale. Fish size is not to scale. RA—relative amplitude.

S3 Table), meaning that as sounds become longer, more pulses are stacked together. On the other hand, the two spectral variables PF and FM were negatively correlated (*Spearman* r = -0.43, n = 67, *P* < 0.001; S3 Table). Later, for the principal component analysis (PCA), we used five acoustic variables (SR, DUR, PRR, PF and FM) which were not highly intercorrelated (r < 0.5). In the PCA performed without the size correction on acoustic variables, the axes PC1 and PC2 accounted cumulatively for (62.5%) of the variation, explaining 40.2 and 22.3% of the variance, respectively (S4 Table). Acoustic properties of the sound FM (positively), PRR (positively) and PF (negatively) were associated with PC1, while sound variable DUR negatively contributed to PC2 (S4 Table). The PC1 *versus* PC2 scatterplot of the taxa illustrated the acoustic variation between gobiids according to sound properties (S1A Fig). After we eliminated the effect of TL on sound features and used these corrected values, PCA accounted cumulatively for 80% of variation, with temporal PRR,DUR and SR being strongly (-0.85,-0.82 and -0.81) associated with PC1, while spectral FM and PF (-0.93 and 0.48) were correlated with PC2 (S1B Fig, S4 Table).

Stepwise Discriminant function analysis (sDFA) differentiated the gobies according to their sound properties (Wilks' Lambda = 0.0002, $F_{28,264}$ = 25.28, n = 67, *P* < 0.001). The first two discriminant functions (DF1 and DF2) cumulatively explained 64.4 and 27.5% of the variation, with DF1 significantly loaded with sound properties PRR and PF, while DF2 showed a positive correlation with NP and DUR (Table 2). In addition, the DFA indicated that individual sounds were correctly classified into the corresponding species with an overall 92.5% correct classification rate (Table 3). Accuracy of the classification rate varied among the species, with *P. bonelli*, *G. paganellus*, *G. niger*, *P. kessleri* and *Z. ophiocephalus* classified with an accuracy of 100%, while *G. cobitis* had the lowest fidelity (66.6%), indicating that sounds could be more variable in this species compared to others. However, these lower percentages of correct species classification, such as for *G. cobitis* and P. *nigricans*, could result from smaller number of individual means used in DFA (N < 7). Partial Wilks' Lambda (for all variables < 0.5, *P* < 0.001) indicated that acoustic variables SR, DUR, NP, PRR and PF contributed, in that order, to the overall discrimination, while FM did not contribute (Partial Wilks' Lambda > 0.5, *P* = 0.10). The DFA differentiated several groups of species (Fig 3). Accordingly, *P. kessleri* and *G. paganellus* were clustered on the positive part of DF1 and DF2 scatterplot. *Padogobius bonelli* was plotted on the negative DF1 and on the positive DF2 in comparison with the remaining two species. These two groups were separated from the others due to high PRR, NP and DUR values, which contributed significantly to DF1 and DF2. *Neogobius fluviatilis*, *N. melanostomus* and *P. nigricans* were plotted on the positive part of DF1 and negative part of the DF2 scatterplot, mainly because their sounds are characterized by a short duration with low NP and high PRR. Furthermore, *G. niger* and *G. cobitis* were plotted on the negative part of the DF1/DF2 scatterplot (DF1 factor coordinates to -6) while *Z. ophiocephalus* was separated from the two species due to the negative DF1 (DF1 factor coordinates > -10) (Fig 3). *Gobius niger* and *G. cobitis*, together with *Z. ophiocephalus*, produce long duration sounds with a high NP but low PRR. By extracting the two most important temporal acoustic properties from DFA, NP and PRR, a scatterplot (based on the species means) was built to illustrate the acoustic structure in greater detail (Fig 4). Species producing short tonal sounds low DUR, low NP, high PRR; *N. melanostomus*, *N. fluviatilis* and *P. nigricans*) clustered on the lower right part of the diagram, while the species situated on the lower left part are characterized by pulsatile sounds (*Z. ophiocephalus*, *G. niger* and *G. cobitis*; intermediate DUR, high NP, low PRR). Finally, the two species producing long tonal sounds are positioned in the upper-middle part of the diagram high DUR, high NP, high PRR; *P. kessleri*, *G. paganellus* see Fig 4). *Padogobius bonelli* was isolated in the upper right part of the diagram (high DUR, high NP, high PRR).

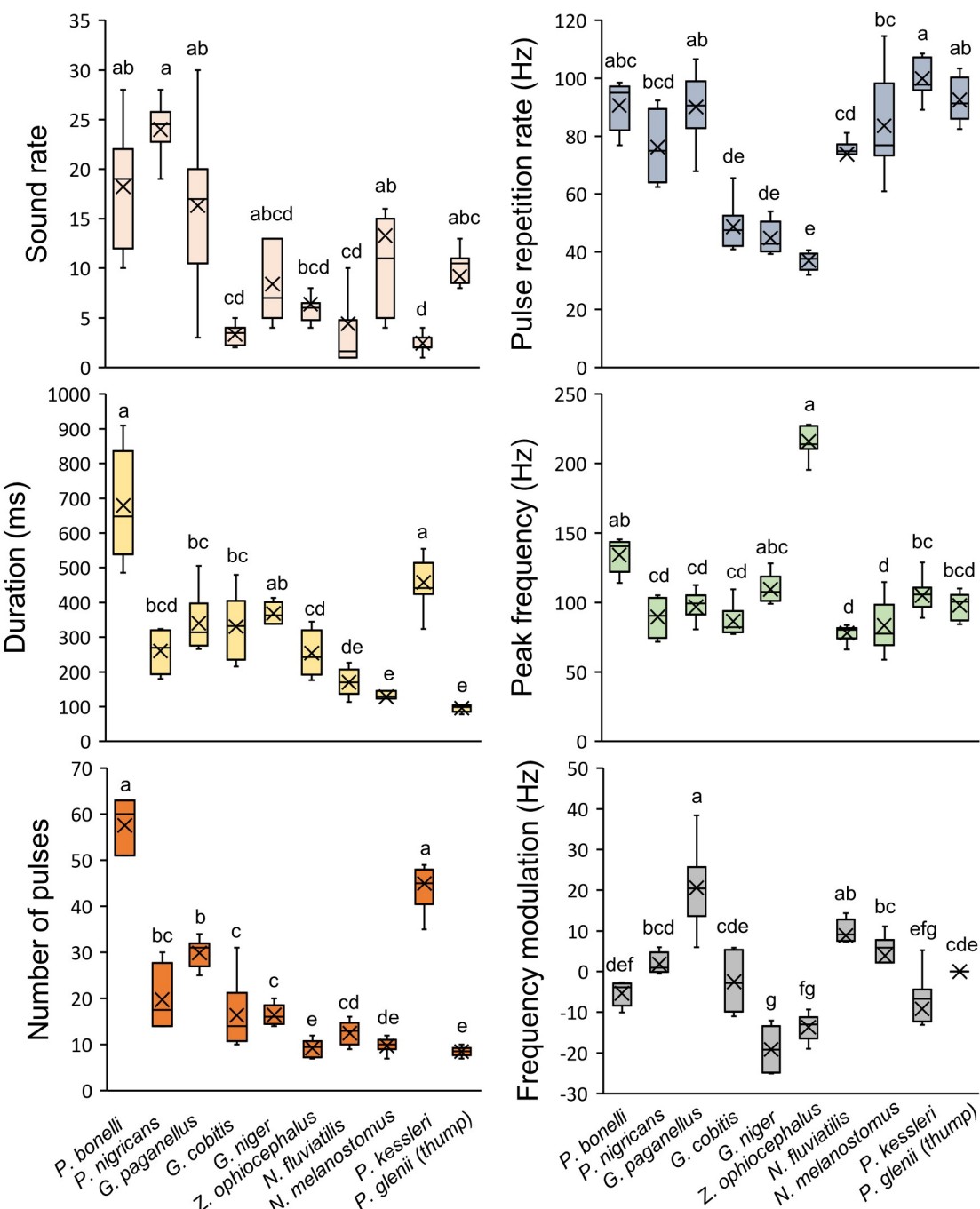

**Fig 2. Box plot of the six acoustic variables of the sounds produced by the investigated gobioid species.** The midline represents the median, x marks the mean, box values indicate the 25th and 75th percentiles, while the whiskers indicate minimum and maximum values of the acoustic properties for each species. Different lowercase letters inside each graph indicate significant differences.

## Phylogenetic affinities between soniferous gobiids

The molecular analysis of a concatenated dataset, inferred from two nuclear (*rag1* & *rho*) and two mitochondrial (*cytb* & *cox1*) molecular markers, allowed us to reconstruct the

**Table 2. Factor structure coefficients from the discriminant function analysis (DFA) representing the correlations between the six acoustic variables and the respective discriminant functions (DF).** In the DFA, species was set as the grouping variable and the individual means of the six acoustic properties as the dependent variables.

| Variable | DF1 | DF2 |
| --- | --- | --- |
| SR (s/min) | 0.05 | 0.06 |
| DUR (ms) | -0.06 | 0.52 |
| NP | 0.19 | 0.91 |
| PRR (Hz) | 0.44 | 0.37 |
| PF (Hz) | -0.41 | 0.21 |
| FM (Hz) | 0.07 | -0.05 |

**Table 3. Stepwise classification matrix indicating the number of cases (individuals) correctly classified in corresponding species or are misclassified according to their acoustic signals.** Total classification rate is also indicated.

| Species | % | 1. | 2. | 3. | 4. | 5. | 6. | 7. | 8. | 9. |
| --- | --- | --- | --- | --- | --- | --- | --- | --- | --- | --- |
| 1. *P. bonelli* | 100 | 5 | 0 | 0 | 0 | 0 | 0 | 0 | 0 | 0 |
| 2. *P. nigricans* | 75.0 | 0 | 3 | 1 | 0 | 0 | 0 | 0 | 0 | 0 |
| 3. *G. paganellus* | 100 | 0 | 0 | 15 | 0 | 0 | 0 | 0 | 0 | 0 |
| 4. *G. cobitis* | 66.7 | 0 | 0 | 1 | 4 | 1 | 0 | 0 | 0 | 0 |
| 5. *G. niger* | 100 | 0 | 0 | 0 | 0 | 5 | 0 | 0 | 0 | 0 |
| 6. *Z. ophiocephalus* | 100 | 0 | 0 | 0 | 0 | 0 | 8 | 0 | 0 | 0 |
| 7. *N. fluviatilis* | 87.5 | 0 | 0 | 0 | 0 | 0 | 0 | 7 | 1 | 0 |
| 8. *N. melanostomus* | 85.7 | 0 | 0 | 0 | 0 | 0 | 0 | 1 | 6 | 0 |
| 9. *P. kessleri* | 100 | 0 | 0 | 0 | 0 | 0 | 0 | 0 | 0 | 9 |
| Total | 92.5 | 5 | 3 | 17 | 4 | 6 | 8 | 8 | 7 | 9 |

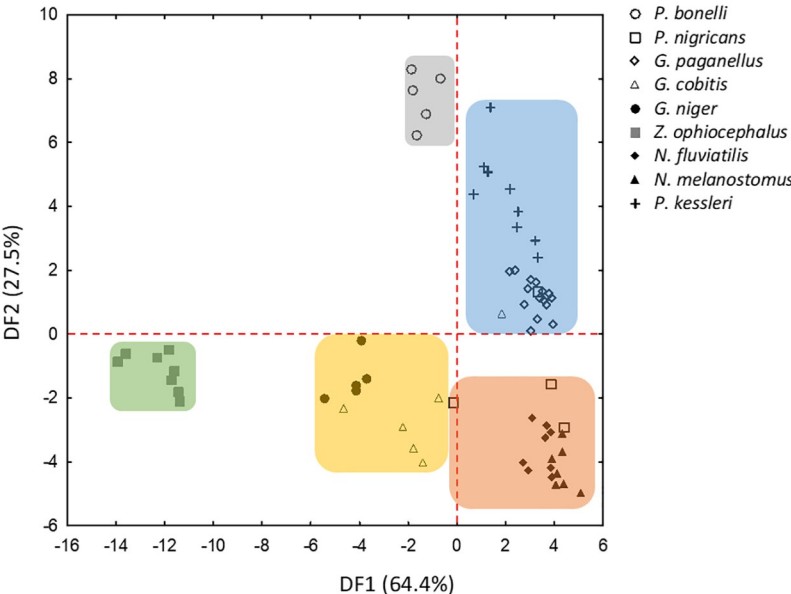

**Fig 3. Scatterplot of discriminant function 1 (DF1) *versus* discriminant function 2 (DF2) performed with individual means of the six acoustic properties from nine gobiid species (*Gobius* lineage).** Each species, set as a grouping variable, is represented by a different symbol. Different colours indicate divergent groups of species according to their acoustic variables.

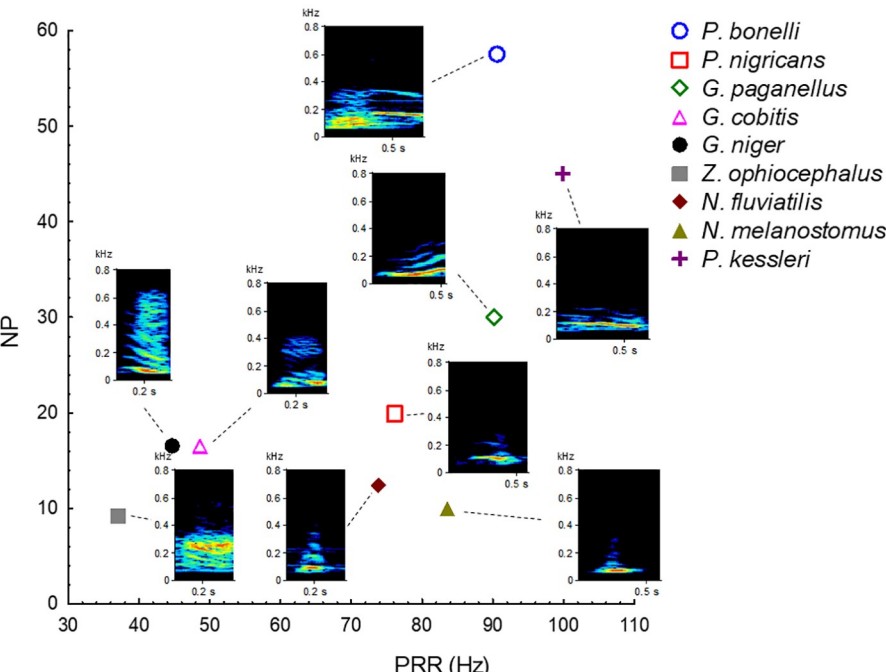

**Fig 4. Categorized scatterplot of two temporal variables (NP *versus* PRR) highlighting the acoustic variability between the nine gobiid species.** For each species, the representative spectrogram in kilohertz is mapped, where brighter colours indicate higher energy intensity. Sounds were recorded at 44.1 kHz and 16-bit resolution while the spectrogram was prepared using AVISOFT software (Hamming window, 512-points FFT, resolution 7 Hz). On the scatterplot, each symbol represents the plot of a selected variable (species mean for NP) against the value of another selected variable (species mean for PRR) broken down (i.e., categorized) by the grouping variable (Species). NP—number of pulses, PRR—pulse repetition rate.

phylogenetic relationships of nine soniferous European gobiids and to build the genetic distance matrix for the pairwise comparison using acoustic data (Table 4). The matrix of 3961 base pairs (bp) contained 30% variable sites, of which 17% are parsimony informative. The sequence lengths of individual molecular markers were: *cytb* 1113, *cox1* 650, *rag1* 1458 and *rho* 740. Phylogenies reconstructed based on the concatenated data using maximum likelihood (ML) and Bayesian inference (BI) method showed identical topologies. *Padogobius bonelli* is in a sister position to all remaining gobiids (Fig 5). This species inhabits

**Table 4. Genetic distance matrix estimated from the concatenated dataset (*cytb*, *cox1*, *rag1* and *rho*) using the p-distance method in MEGA.**

| Species | 1. | 2. | 3. | 4. | 5. | 6. | 7. | 8. |
|---|---|---|---|---|---|---|---|---|
| 1. *G. cobitis* | | | | | | | | |
| 2. *G. paganellus* | 7.73 | | | | | | | |
| 3. *G. niger* | 10.20 | 10.08 | | | | | | |
| 4. *Z. ophiocephalus* | 10.36 | 9.73 | 9.35 | | | | | |
| 5. *P. nigricans* | 9.91 | 10.44 | 11.84 | 10.89 | | | | |
| 6. *P. bonelli* | 10.66 | 11.04 | 12.29 | 11.71 | 11.21 | | | |
| 7. *N. fluviatilis* | 9.40 | 9.86 | 10.98 | 11.06 | 6.20 | 10.86 | | |
| 8. *N. melanostomus* | 9.61 | 9.94 | 11.67 | 11.07 | 6.05 | 11.24 | 5.75 | |
| 9. *P. kessleri* | 10.93 | 11.54 | 12.52 | 12.49 | 10.04 | 12.04 | 9.50 | 9.94 |

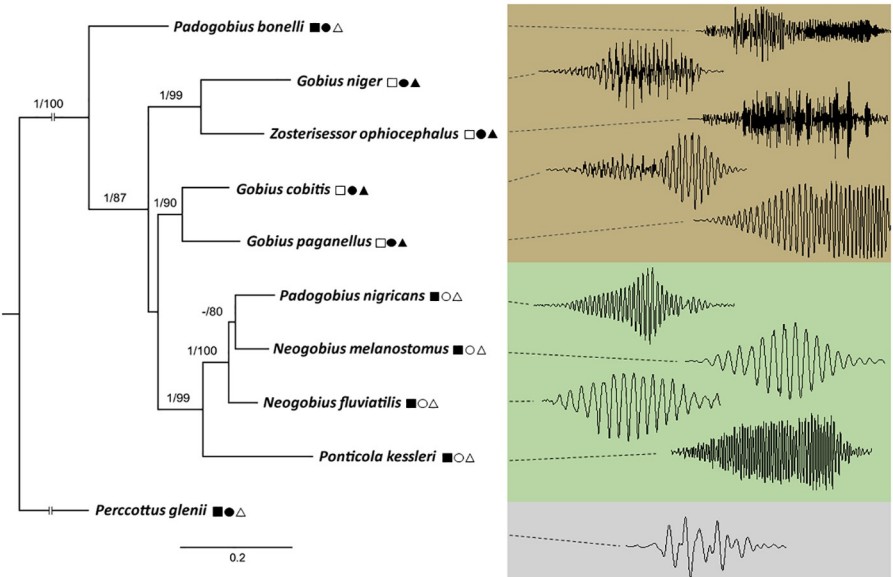

**Fig 5. Bayesian inference phylogenetic relationships between the studied goby species based on concatenated dataset of two mitochondrial (*cytb* and *cox1*) and two nuclear markers (*rag1* and *rho*).** The numbers on nodes represent posterior probability (BI) and bootstrap support (%, ML) values. Nodes with values ≥0.95 for posterior probability and ≥75% for bootstrap support are considered well supported and are depicted;–indicates posterior probability value <0.95. For each taxon, a single representative sound waveform was mapped to underline the acoustic affinities between the investigated taxa. Species groups are distinctly coloured: brown- Atlantic-Mediterranean gobiids; green- Ponto-Caspian taxa; light grey—odontobutid *Perccottus glenii*. Sounds were recorded at 44.1 kHz and 16-bit. In addition, for each species, three morphological or habitat traits are indicated by □ (number of vertebrae: white—less than 28; black—more than 28), ○ (swim bladder: white—absent; black—present) and △ (habitat: white—freshwater; black—marine). Habitat refers to the water type from which individuals for the analysis were captured. Waveforms are not to scale. Branch length scale represents number of substitutions per site.

freshwaters, has a higher number of vertebrae (> 28) and possesses a swim bladder. The remaining species formed three well supported topological groups. Of the four marine gobies (which share the reduced number of vertebrae [< 28] and the presence of swim bladder), *Gobius niger* and *Zosterisessor ophiocephalus* grouped into the first clade, while *G. cobitis* and *G. paganellus* clustered into the second. The third group was composed of gobiids distributed in the Ponto-Caspian region (genera *Neogobius* and *Ponticola*) and *P. nigricans*, an Italian freshwater endemic species (Fig 5). This group occupies freshwater habitats, has a higher number of vertebrae (> 28) and lacks a swim bladder. Specifically, in the third group, *P. kessleri* is a sister taxon to the remaining species, while within the "*Neogobius* group", *Neogobius fluviatilis* is a sister taxon in regards to two closely related species, *P. nigricans* and *N. melanostomus* (Fig 5). These results support the monophyly of these soniferous Ponto-Caspian species (plus *P. nigricans*) and suggest the polyphyly of the genus *Padogobius*. In Mesquite, the Trace Character History method depicted a history of character evolution on the phylogenetic tree, and to reconstruct the ancestral state of categorical character "Sound type", we used this method combined with the Parsimony reconstruction method. In combination, these methods produced a single rooted phylogenetic cladogram from the previously constructed BI phylogenetic tree (Fig 6). The states of the categorical character (i.e., different sound types) were mapped onto this BI phylogenetic tree, and within the parsimony reconstruction (unordered model), relevant statistical measures were calculated (Parsimony tree-length = 6; Consistency index = 0.83; Retention index = 0.50).

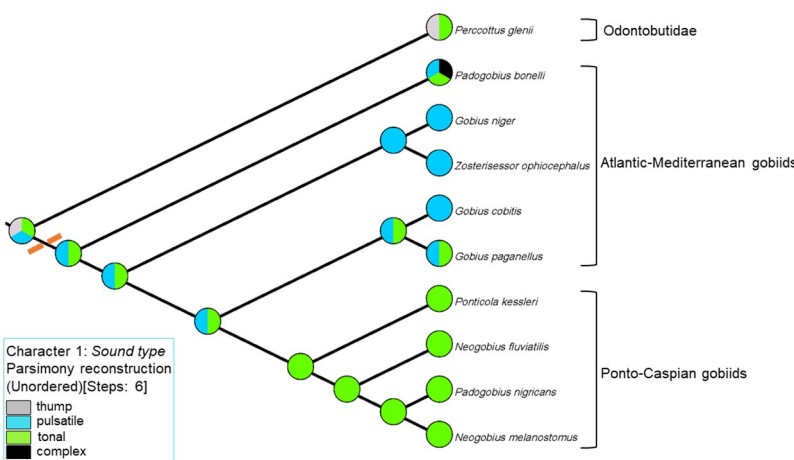

**Fig 6. Cladogram depicting the evolution of acoustic signals and their ancestral states in soniferous *Gobius* lineage gobiids, using *P. glenii* as an outgroup.** Codes for the categorical character "Sound type" were 0—thump; 1—pulsatile; 2—tonal; 3—complex. The dashed orange line indicates the *Gobius* lineage.

### Acoustic and genetic divergence comparison

We performed pairwise comparison between sound divergence (derived from the representative sound types) and genetic diversity to investigate their mutual relatedness in gobiids. Specifically, sound divergence matrix was built from Cluster analysis (Joining tree analysis) in STATISTICA® (clustering performed by using six acoustic features), and genetic distance matrix, which was obtained by using the uncorrected p-distance method based on the concatenated dataset for all used molecular markers (genes *cytb*, *cox1*, *rag1* and *rho*), and separately for mtDNA (*cytb* and *cox1*) and nDNA markers (*rag1* and *rho*). A significant positive correlation was found between acoustic and genetic distance matrices (Mantel test $r = 0.470$, $Z = 2298.756$, $P_{two-tailed} = 0.01$; Fig 7), even after performing the matrix permutation test (10,000 repetitions, $P_{two-tailed} = 0.03$) indicating that in soniferous gobiids from the *Gobius* lineage, interspecific divergence in sound follows the same phylogenetic pattern of diversification. When the acoustic distance was compared with genetic divergence based on nuclear (*rag1* and *rho*) or mitochondrial (*cytb* and *cox1*) molecular markers, the results were similar to those with the concatenated dataset. There was a significant positive correlation between acoustic and nuclear distance (Mantel test for nDNA: r = 0.485, Z = 793.852, $P_{two-tailed} = 0.005$; S2A Fig), and acoustic and mitochondrial distance (Mantel test for mtDNA: r = 0.450, Z = 4195.402, $P_{two-tailed} = 0.01$; S2B Fig), indicating once again that for each type of molecular marker used in this study (mt or nuclear) and its evolutionary rate (faster mtDNA or slower nuclear nDNA), sounds diverged in a similar pattern to the phylogenetic affinities in gobiids.

### Discussion

In comparison with the vocalisations of other vertebrates like frogs, birds or mammals, the relative simplicity and strong stereotypy of teleost sounds make fish a useful group for studying acoustic evolution and its association with phylogeny. By correlating acoustic variability (derived from the representative sound types) with genetic divergence (concatenated or mitochondrial/nuclear genes), we sought to elucidate whether these representative sounds in soniferous gobiids (*Gobius* lineage) have a phylogenetic basis, and discuss whether stochastic evolutionary forces play a prominent role in signal divergence. No similar investigation has ever been performed in gobiids or any other teleost group to date, and our results could be

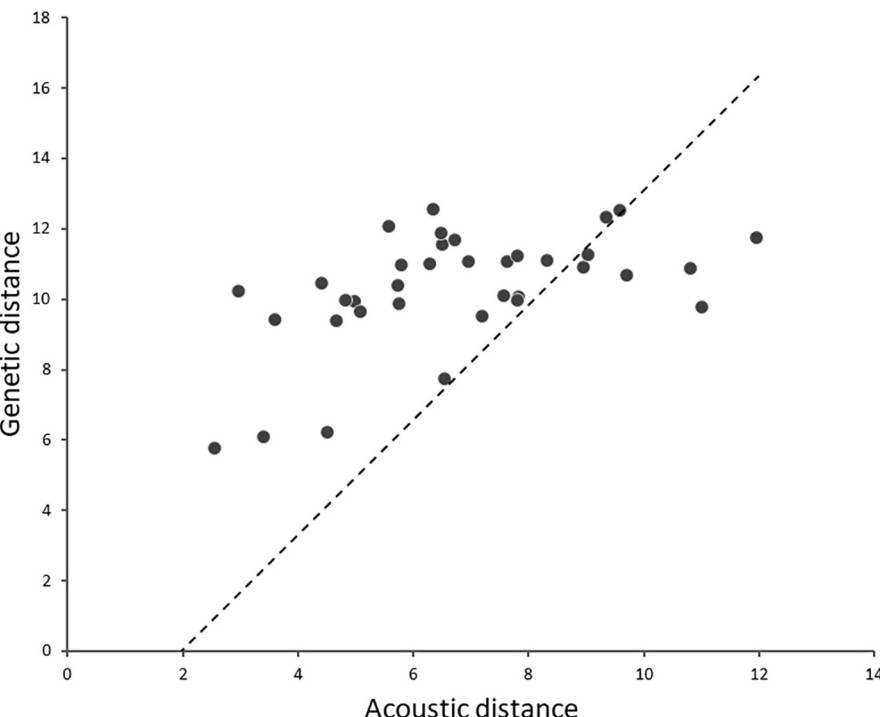

**Fig 7. Correlation between genetic distance and acoustic divergence in nine soniferous *Gobius* lineage gobiids (Mantel test r = 0.47, $P_{t.t.}$ = 0.01).** Genetic distance was estimated from the concatenated dataset (p-distance method), while acoustic distance was estimated from the standardized Manhattan distance metric procedure using species means of the six sound variables for each representative sound type. The dashed line shows the linear trend pattern, while the scatterplot represents the relationship between species genetic differentiation and their acoustic distance. The plots are the coordinates of the relationship.

meaningful, since speciation in animals may be accelerated by the separation of signalling systems [20]. Since various communication signals, including acoustic signals, reveal the identity of the signalling animal, they may be involved in species diversification [11].

### Acoustic variability and phylogenetic relationships between soniferous gobiids

The present acoustic analysis allowed us to discriminate between nine gobiids according to the spectro-temporal properties of their representative sounds, indicating that each taxon produces species-specific acoustic signals characterized by a unique set of variables allowing for interspecific differentiation. As mentioned earlier, for each species the representative sound type was selected (especially in taxa having the ability to produce two or even three different types) and the sounds were extracted from their observed frequency in the behavioural experiments. Here we have shown that even with this incomplete dataset, the acoustic diversity follows the pattern of genetic divergence. Likewise, since most gobies live in natural sympatry with at least one other goby species [57–60], we believe that these sounds could act as accurate species-discrimination traits. From the acoustic analyses including PCA and sDFA, certain sound properties, especially temporal NP, PRR and DUR and spectral PF, appear to be responsible for the observed interspecific acoustic divergence. Since these sound properties accounted for the most variation among species in both PCA and sDFA (variable-factor correlations ranged from 0.4 to 0.9), they can be regarded as the main acoustic components carrying the

phylogenetic signal. The overall similarity in taxa composition was achieved by comparing the sDFA scatterplot with the DUR *versus* NP diagram, and both highlighted a similar pattern of acoustic divergence among the studied gobiids. Moreover, the sDFA emphasized that the individual sounds were accurately attributed to corresponding species with high overall classification fidelity (> 90%), meaning that representative acoustic signals could indeed reflect a phylogenetic taxon affiliation. It is important to note that the number of available soniferous individuals across the species, used in PCA and sDFA as means for each acoustic property, was not homogeneous. However, for each species in our analysis more than 60% of individuals were correctly classified into the appropriate taxon, allowing us to conclude that acoustic signals could be used for phylogenetic purposes. The first comparative study on acoustic signals produced by Mediterranean gobies [47] proposed that relationships between soniferous gobies could be inferred using the signal structure as a reliable indicator of taxon affiliation, given the strong relationships between acoustic affinities and species traits. However, without genetic data, this was a long-standing hypothesis corroborated by the present study. Our phylogenetic analysis strongly indicated that *P. bonelli* is separated from the rest of the investigated taxa, occupying an isolated position on both the sDFA phenogram and concatenated phylogenetic tree. Among the Atlantic-Mediterranean gobiids (*Gobius* and *Zosterisessor* genera), deeper phylogenetic relationships remain unresolved, though some interesting observations can be drawn from our results. For *Z. ophiocephalus* and *G. niger*, an isolated group on the genetic tree, the observed phylogenetic relationship coincides with their habitat preferences [mud or silt; 59]. Likewise, their close phylogenetic affiliation has been confirmed using DNA sequences from both mitochondrial and nuclear molecular markers [42, 43, 45], although these studies included a smaller number of species for phylogenetic analysis. In the PCA and sDFA, *Z. ophiocephalus* had a more isolated position in both PCA and sDFA scatterplot compared to the *G. cobitis* + *G. niger* group, likely due to the higher spectral sound properties, probably PF. These observations shed light on the taxonomic position of *Z. ophiocephalus*, as some authors have suggested a close phylogenetic relationship with the genus *Gobius*, while others still isolate *Z. ophiocephalus* into a monotypic genus. While these doubts remain to be clarified by future studies, for the moment, our study indicates that phylogenetic relationships obtained from sounds and molecular markers, are highly complex in *Zosterisessor* and *Gobius* species. On the other hand, *G. cobitis* and *G. paganellus* are mostly found on rocky bottoms, and they share certain phenotypic traits [e.g., colouration pattern and sagittal otolith shape; 59, 91]. We hypothesize that the acoustic similarity between these two species would have been stronger if we included the additional sound type (i.e. pulsatile sound) recorded from *G. paganellus* in the acoustic analysis. However, this was not possible in the present study, and therefore, acoustic differences between these two species appear higher than their genetic differences. The species inhabiting the Ponto-Caspian region (genera *Neogobius* and *Ponticola*), produce only one sound type [33, 48], justifying their clustering into one well-supported clade [this study and 42, 43, 46, 92]. However, from our results, a certain degree of acoustic variability is evident between the *Neogobius* group (*N. fluviatilis*, *N. melanostomus* and *P. nigricans*), characterized by the production of short tonal sounds, and *P. kessleri*, which was separated from the remaining Ponto-Caspian species in the PCA and sDFA diagrams, likely due to its long, frequency-modulated sounds, similar to *G. paganellus* acoustic signals. This is interesting since ecologically, both species (together with *P. bonelli*) share similar bottom preferences (rocks or coarse gravel), although *P. kessleri* is a freshwater resident. This observation might suggest that tonal sounds are more suitable for hard-bottom transmission, although this should be examined in future studies by investigating the ecological adaptations of gobies to sound production and certain habitat conditions. Recent studies have suggested that rocky or pebbly substrates, inhabited by the bottom-dwelling gobies producing tonal sounds (such as

*G. paganellus*, *P. kessleri* and *P. bonelli*), are unfavourable for sound emission, due to the low-frequency ambient noise and short-range transmission of sounds [93, 94]. [95] proposed that tonal sounds could possess characteristics enabling longer-range transmission than pulsatile sounds, since the acoustic structure is simpler. In addition, according to [95], waveform differences between pressure and particle velocity spectra are less expressed in tonal than pulsatile sounds. In vertebrates, frequency modulated sounds (such as the tonal sounds of gobiids) are generally long-range signals [96]. Furthermore, the close affinity of *P. nigricans*, an Italian endemic goby, with the Ponto-Caspian group in both the PCA, sDFA and the phylogenetic tree is phylogenetically interesting. Morphologically, according to [57], *P. nigricans* is similar to the Ponto-Caspian gobies due to the higher number of vertebrae (> 29), absence of swim bladder and the presence of head canals, while *P. bonelli* has a swim bladder but lacks head canals [57]. Likewise, recent acoustic studies indicated that *Padogobius* could be polyphyletic [33, 48], supporting previous molecular findings that emphasized that the two *Padogobius* species are of independent origin [68, 97, 98]. However, no studies have used representative acoustic datasets to investigate sound diversity and to combine genetic data with sound variability to empirically confirm this hypothesis. Our comparative study strongly corroborated these hypotheses, indicating that the genus *Padogobius* is truly polyphyletic. Acoustic signals, as shown here, even when using representative sounds, proved to be a valuable species-specific trait in the European gobiids, and a suitable basis for future phylogenetic studies. It is important to note that of the overall number of soniferous gobiids, the sounds of *Proterorhinus marmoratus* (Pallas, 1814) [99] and *Gobius cruentatus* Gmelin, 1789 [100] were not included in the present analysis due to technical reasons, but their ability for sound production offers the possibility for future interspecific studies.

## Evolutionary forces driving acoustic divergence

In young or emerging species, acoustic signals may serve as isolating mechanisms, leading to intraspecific acoustic variability. It has long been debated whether selection or drift have relative importance in the process of speciation [101, 102]. To demonstrate that stochastic ("neutral"), and not deterministic ("adaptive") evolution is most important in driving acoustic differences, divergence in acoustic traits should be empirically confirmed to increase linearly with genetic distance, with little or no effect of selection [11, 20]. Although we did not test the effects of selection, a positive linear correlation between acoustic distance and genetic divergence for all investigated species was obtained, using both a concatenated molecular dataset and individual mtDNA/nDNA molecular markers. Generally, it was assumed that mtDNA evolves at a faster rate than nDNA in animals, and the study carried out by [103] proposed that the ratio of the mtDNA to nDNA mutation rate is around 12:1 in Teleostei. Our results indicate that sounds diverge in the same manner as DNA sequences, irrespective of the molecular marker type (mitochondrial or nuclear) and its evolutionary rate (faster mtDNA and slower nDNA). In the present study, the coefficient of correlation between acoustic and genetic divergences (concatenated or mitochondrial/nuclear) was relatively weak and ranged from 0.45 to 0.47. This suggests that other forces, probably deterministic, could have an additional role in acoustic divergence. Generally, in the presence of a positive correlation, most studies have emphasized that drift could be the main driver behind differentiation in sounds between species [11, 104, 105]. However, we propose that this should be verified by future studies exploring the intensity of sexual selection between closely related species which, if acoustic distance does not correlate with genetic, would then justify our weaker Mantel correlation values. Other factors such as social learning [106] and mutation-order processes [107] could also drive acoustic differentiation. In order to prove that drift, social learning or mutation-order processes play a

role in acoustic divergence, some preconditions must be met. In learned vocal signals [e.g., bird song, 108; whale song, 109], cultural transmission and copying errors are major drivers of stochastic divergence within soniferous populations. The ability of gobiids to learn sounds has never been investigated. Considering the recent findings of other teleosts, suggesting that the sounds in fish are innate [110, 111], the effects of social selection as a driving force for the observed divergence can be excluded at this time. Likewise, mutation-order processes over time can cause the linear accumulation of acoustic differences [112, 113], resulting in a highly complex interaction between such processes and drift. However, some gobiids hybridise [114], violating reproductive isolation as one of the main criteria for mutation-order speciation. Therefore, at this time, the sole empirical evidence that the present study offers is the positive correlation between acoustic and genetic distances. As mentioned above, the divergence may result from a combination of selection and drift [20, 115], which would in fact, corroborate our observed (weaker) correlation between acoustic and genetic distance.

It appears from the positive correlation that the acoustic signals (presented here as representative sound types for each species), including their acoustic features, could carry important phylogenetic signal for species recognition. However, the exact degree of the phylogenetic signal carried by the acoustic features of these representative gobiid sounds and their rate of evolution remains unclear. In their elaborate study, [113] tested whether differences in a male sexual signal (nuptial colour) were correlated with environmental, genetic or geographic distances in darters (Percidae). From the observed correlation between overall male colour differences (i.e., scores for discrete colour categories) and genetic divergence, they concluded that a single phenotypic trait, i.e., breeding coloration of males (or in our case sound), could possibly be a combination of various independent (continuous and discrete) characters, each operating under different selective regime [113]. With this in mind, we can expect that certain sound properties (NP, DUR, PRR and PF) would carry different levels of phylogenetic information.

## Hypothesis explaining acoustic divergence in gobiids

According to the results of the Trace Character History method from Mesquite (Fig 6) and the existing literature, we were able to reconstruct the hypothetical scenario explaining the evolution of acoustic signals in soniferous gobiids presented here, along with other soniferous gobioids (Fig 8). It is important to note that for some gobiiform groups, such as Rhyacichthyidae, Butidae or Eleotridae, there is a complete gap in the knowledge about their acoustic abilities, and therefore, this hypothesis is still hypothetical for groups outside the *Gobius* lineage. Briefly, three different sounds types [thumps (i.e., short irregular pulses), tonal and pulsatile] could represent the hypothetical ancestral state, i.e., the symplesiomorphic condition, for gobioids. Two of these, thumps and tonal, are present in the basal gobioid *P. glenii*, while the pulsatile signals were recorded from another basal species, *O. obscura* [116]. This situation is justifying the observed character state (i.e., three sound types) at the basal node of the cladogram (Fig 6). Within the *Gobius* lineage, tonal and pulsatile sounds were maintained as ancestral state characteristics, while the complex sounds (composed from pulsatile and tonal sound types) constitute a autapomorphic trait among our investigated species that has evolved in *P. bonelli*. For *Gobius* genus (including *Z. ophiocephalus*), both pulsatile and tonal sounds were the ancestral state, while this was further simplified in the terminal taxa group (node *G. niger* + *Z. ophiocephalus*) where only pulsatile sounds are documented. Lastly, the complete Ponto-Caspian group (together with *P. nigricans*) shares the single ancestral character state (i.e., tonal sounds), which is present at the internal node and in all the terminal taxa presented here. In the sister group to Gobiidae, the sand gobies (Gobionellidae), only pulsatile and thumps sounds have been recorded to date [8]. In combination with the known acoustic diversity in

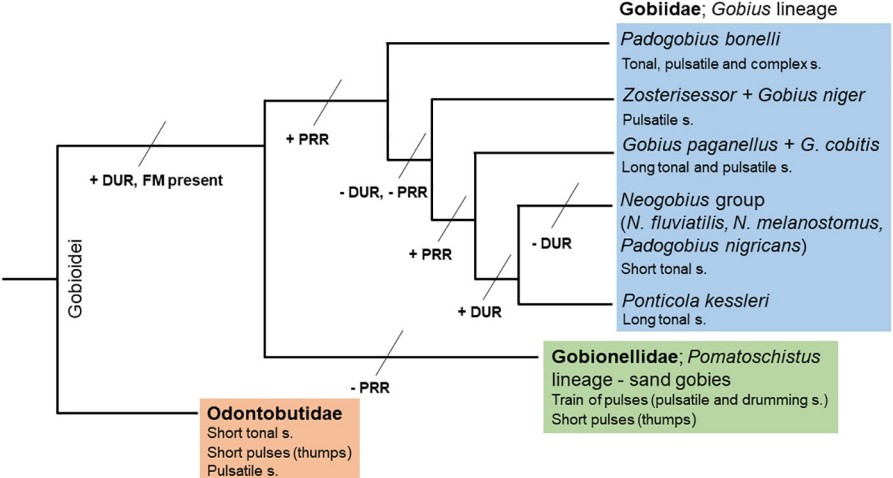

**Fig 8. Diagram depicting the divergence of acoustic signals between soniferous gobioids following the evolutionary hypothesis.** Relationships between the gobiids follow the interspecific relationships obtained from the concatenated Bayesian inference (BI) phylogenetic tree from the present study. Abbreviations: DUR—duration (ms), PRR—pulse repetition rate, FM—frequency modulation (Hz), s., sounds.

odontobutids, these results confirm that the largest acoustic diversity is presently documented in the Odontobutidae and *Gobius* lineage gobiids.

## Conclusions

In summary, our results suggest that each species of the soniferous European gobiids examined here could be recognized based on its acoustic structure and spectro-temporal features of its representative sounds, making the acoustic signals a highly species-specific trait. Acoustic and genetic analyses recognized several species groups, and since *P. nigricans* clustered in the same acoustic and genetic topology with *Neogobius* spp., we suggest that the genus *Padogobius* is polyphyletic and that *P. nigricans* is closely related to the Ponto-Caspian gobies. Furthermore, our comparative acoustic-genetic analyses explored the pattern of sound divergence, which correlated linearly with genetic distance. Therefore, we propose that certain acoustic properties of gobiid sounds carry a phylogenetic signal responsible for species recognition. In conclusion, we strongly suggest that sounds in European gobiids represent a promising phylogenetic tool for future comparative studies aiming to resolve their affinities and taxonomic status.

## Supporting information

**S1 Fig.** Scatterplot from principle component analysis (PCA) performed with individuals means of the five acoustic variables from nine gobiid species (*Gobius* lineage), performed A) without the correction on acoustic variable for size and B) with correction for size ("$XTL^{-1}$", where "$X$" is the acoustic variable). In A) PC1 is loaded by variables frequency modulation, pulse repetition rate and peak frequency, while PC2 by duration. In B) PC1 is loaded with pulse repetition rate, duration and sound rate, while PC2 by frequency modulation and peak frequency.
(TIF)

**S2 Fig. Correlation between genetic distance and acoustic divergence in nine vocal *Gobius* lineage gobiids.** In A), correlation was achieved (Mantel test r = 0.48, $P_{t.t.}$ = 0.005) by using genetic distance obtained from p-distance method for nuclear markers (*rag1* and *rho*), while

in B), correlation (Mantel test r = 0.45, $P_{\text{t.t.}}$ = 0.01) was inferred from mitochondrial *cytb* and *cox1* sequences while the divergence was obtained using p-distance method. The scatterplot represents the relationship between species genetic differentiation and their acoustic distance.
(TIF)

**S1 Table. Mean values and standard deviations of the total length and the six acoustic variables for the ten gobioid species.** For *Perccottus glenii*, only thump sounds were used for the acoustic analysis, for which FM could not be calculated (/). Number of recorded individuals per species (N) and number of analysed sounds (n) are indicated. Abbreviations: TL—total length, SR—sound rate, DUR—duration, NP—number of pulses, PRR—pulse repetition rate, PF—peak frequency, FM—frequency modulation.
(PDF)

**S2 Table. Partitioning scheme and best-fit models of evolution for data blocks defined by gene and codon position, assessed by PartitionFinder 2 for subsequent phylogenetic analyses in MrBayes and RAxML.**
(PDF)

**S3 Table. *Spearman* correlation coefficient of the relationships between the six acoustic properties.** Correlation is based on the individual means of six acoustic properties per species (N = 9).
(PDF)

**S4 Table. Percentage and cumulative percentage of variance explained by the first two axis of principal component analysis (PCA), with the loadings for these axes (i.e., factor coordinates) extracted from five acoustic variables from nine gobiid species (*Gobius* lineage).** PC factor coordinates represent the correlations between the respective individual mean value of sound variable and each PC factor. PC (without) indicate the percentages and loadings of acoustic variables obtained without the correction for size, while PC (with) highlights the percentages and loadings obtained with the acoustic variables corrected for size ("$XTL^{-1}$", where "*X*" is the acoustic variable). For PCAs, we excluded acoustic variable number of pulses (NP) due to its correlation with other variable (DUR).
(PDF)

**S1 Data.**
(XLSX)

## Acknowledgments

The authors are grateful to technician S. Vajdić and R. Karlović for their dedicated help during fish sampling. M. Kovačić provided the genetic material for the two marine species (G. *cobitis* and *G. paganellus*), while M. Smederevac-Lalić and S. Skorić provided the live specimens of *Perccottus glenii*. Finally, we thank I. Bacelj for the help with statistical issues and L. Zanella (WindWord; www.windword.hr) for the professional proofreading of this paper.

## Author Contributions

**Conceptualization:** Sven Horvatić, Stefano Malavasi.

**Data curation:** Sven Horvatić.

**Formal analysis:** Sven Horvatić, Stefano Malavasi, Jasna Vukić, Ivana Buj, Lucija Onorato, Lucija Ivić.

**Funding acquisition:** Jasna Vukić, Davor Zanella.

**Investigation:** Sven Horvatić, Stefano Malavasi, Jasna Vukić.

**Methodology:** Sven Horvatić, Stefano Malavasi, Jasna Vukić, Radek Šanda, Zoran Marčić, Marko Ćaleta, Massimo Lorenzoni, Perica Mustafić, Francesco Cavraro, Davor Zanella.

**Project administration:** Sven Horvatić, Jasna Vukić.

**Software:** Sven Horvatić, Jasna Vukić.

**Supervision:** Stefano Malavasi, Davor Zanella.

**Validation:** Sven Horvatić, Stefano Malavasi, Jasna Vukić.

**Visualization:** Sven Horvatić, Perica Mustafić.

**Writing – original draft:** Sven Horvatić, Jasna Vukić.

**Writing – review & editing:** Sven Horvatić, Stefano Malavasi, Jasna Vukić, Radek Šanda, Zoran Marčić, Marko Ćaleta, Massimo Lorenzoni, Perica Mustafić, Ivana Buj, Lucija Onorato, Lucija Ivić, Francesco Cavraro, Davor Zanella.

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
