## [Decision Letter · Decision Letter 0]

14 Jun 2021

PONE-D-21-00246

Correlation between acoustic divergence and phylogenetic distance in vocal European gobiids (Gobiidae; Gobius lineage)

PLOS ONE

Dear Dr. Zanella,

Thank you for submitting your manuscript to PLOS ONE and apologies for the delay in the decision. It took a long time to find reviewers who agreed and I was also waiting on a reviewer who stopped all communication.

After careful consideration, we feel that it has merit but does not fully meet PLOS ONE’s publication criteria as it currently stands. Therefore, we invite you to submit a revised version of the manuscript that addresses the points raised during the review process.

In particular, it would be important to address the comments made by reviewer 2 about prviding further details and definitions in the manuscript.

We look forward to receiving your revised manuscript.

Kind regards,

Vivek Nityananda

Academic Editor

PLOS ONE

Journal Requirements:

1.Please ensure that your manuscript meets PLOS ONE's style requirements, including those for file naming. The PLOS ONE style templates can be found at and

Reviewers' comments:

Reviewer's Responses to Questions

**Comments to the Author**

1. Is the manuscript technically sound, and do the data support the conclusions?

Reviewer #1: Yes

Reviewer #2: Partly

2. Has the statistical analysis been performed appropriately and rigorously? 

Reviewer #1: Yes

Reviewer #2: Yes

3. Have the authors made all data underlying the findings in their manuscript fully available?

Reviewer #1: Yes

Reviewer #2: No

4. Is the manuscript presented in an intelligible fashion and written in standard English?

Reviewer #1: Yes

Reviewer #2: No

5. Review Comments to the Author

Reviewer #1: The authors obtained solid data to support their main conclusions. The overall writing of this manuscript is good except for few typos or errors (such as Lines 23 & 42: "Teleost" should be "teleost"; Lines 220: "implies" should be "imply").

Reviewer #2: The present manuscript from Horvatić et al., investigates the relationship between the acoustic variability and the phylogenetic distance of sonic teleost fish, European gobiids. The authors analysed the spectro-temporal acoustic properties of nine vocal gobiids from pre-existing audio materials recorded in other studies to assess their variability. They constructed a phylogenetic relationship between these nine sonic gobiids based on concatenated genetic sequences from mitochondrial and nuclear genes. Finally, the authors confronted the genetic distance with the acoustic divergence to assess whether the degree of correlation could explain the hypothesis that interspecies sound variability falls under stochasticity.

Overall, the hypothesis developed in this manuscript is interesting and the detailed procedure provided in the method section is appreciated. However, the study would benefit of additional evidences to support the discussion and conclusions. The terminology of bioacoustics and evolutionary processes must be appropriately defined and used to avoid misinterpretation of the concepts developed here. Some reorganisations in the discussion, corrections and precisions throughout the manuscript, as well as moderation of statements are suggested below. The manuscript would benefit from proofreading for consistency.

Abstract

Line 20. Is identity related to individual or species acoustic identity? Please precise.

Line 30. Acoustic communication should be replaced by acoustic signals. No behavioural experiment was conducted in this study; see related comment below in the Method section - sound recordings and analyses.

Line 36. Please, rephrase to better discriminate between what was “invoked” by others studies versus what is corroborated in the present study.

Introduction section

Line 56, 57, 62. Why using alternatively double and single inverted coma? Please homogenise throughout the manuscript if no specific significance.

Line 68. Typo: support instead of supporting.

Line 69. If “patterns” refer to acoustic variation and genetic divergence please replace by a more global and inclusive word, e.g. notions, components…

Line 71-76. Please consider rearranging/rephrasing: The vocal repertoire among Teleost fishes has been thoroughly investigated. In the gobioids (Gobiiformes; Gobioidei), soniferous species produce different types of acoustic signals consisting of a rich repertoire composed of pulsatile and tonal sounds. This acoustic repertoire shows a great variability at both the inter- and intraspecific levels, with four different sound types [thump, pulsatile (drum), tonal, and complex] recorded to date, emitted mainly by males as part of the breeding and aggressive sonic behaviours.

Line 73-76: Are females not able to produce sounds in any of the cited behavioural contexts, or in general?

Line 76. British English: behavioural; please homogenise throughout the manuscript.

Line 77. Redundant, choose either one: …like in, for example,…

Line 80-81: American English: specialization, utilize; please homogenise throughout the manuscript.

Line 80. Please quickly explain what “lack of sonic specializations” refers to.

Line 81. “currently”, confusing and may be removed.

Line 82. Note: Vocal communication commonly refers to sounds produced with respiratory system (the larynx in mammals and reptiles or the syrinx in birds). Some acoustic studies in fish use that term to define precise sound types often associated with mechanisms involving air pathways as the swim bladder. Thus, vocal communication, vocalisation and call terms should be wisely used or must be replaced by more general terms, e.g. sonic or acoustic communication, as the present manuscript refers to various sound types from species sometimes lacking the swim bladder.

Line 83-100. Please reorganise this paragraph to facilitate the reading.

Replace or precise “species-rich groups” with the appropriate and consistent terminology, e.g. when putting into opposition “[…] Gobius-lineage form a monophyletic group […] whereas the genus Gobius […] the most species-rich groups.”

Line 93-100. The description of the percentage and proportion of sonic species in each group and lineages is difficult to follow. Similarly, the authors are describing and citing published studies in a way that one could understand they are describing experiments performed in the present study.

Line 93. Are experimental trials referring to behavioural trial? If so, please quickly precise or replace experimental by behavioural.

Line 96. “vocally”, please see comment above, line 82.

Line 97. Are bioacoustics experiments trials referring to behavioural trial? Same comment.

Note. Please make sure to discriminate between the capability of fish to produce sounds and acoustic communication per se and to use each concept in a more accurate manner.

Line 103. to a certain degree, please modify.

Line 102. “call” please see comment above, line 82.

Line 105. “other fish groups”. A little vague, replace or precise “groups” with the appropriate and consistent terminology.

Line 111. American English: hybridization.

Line 111. “vocal” please see comment above, line 82.

Line 112. American English: specialization.

Line 114. “genetic markers”. Precise to what the term marker refers to here or replace, or remove.

Line 117. “in vocal Gobius lineage gobies”. Redundancy, gobies should probably be removed.

Line 120. “multiple genetic markers”. Same comment as above, line 114. The authors also may want to elaborate in the introduction which genetic markers they chose and the rational for choosing two mitochondrial and two nuclear genes, how and why they were chosen?

Line 129-131. Note. Points iv and v may be moderated and perhaps rethought and rephrased in the light of the comments in the Discussion section (see Discussion section).

Line 131. “acoustic communication”. Please see comment above, line 82.

Material and method section

Paragraph: Study species (Replace by Studied species)

Line 135-140: Please, remove and reorganise with less redundancy with other paragraphs of this section and for a clearer understanding of the rational for choosing the nine species as suggested hereafter: This study analysed acoustic signals and our species composition was based on the availability of audio tracks enabling the comparison of nine soniferous gobiids (Gobiidae, Gobius lineage) belonging to five genera (Gobius, Padogobius, Zosterisessor, Neogobius & Ponticola). Amur sleeper […]

Line137-139. “The sounds were previously recorded and described by the authors of the present study (see ‘Sound recording and bioacoustic analyses’) …” This statement could be introduced in the acoustic recording and bioanalysis paragraph, with the reference of the original study where the audio material was recorded from and/or already used.

Line 151. Does “per. Obs.” Means personal observations?

This paragraph would benefit the addition of the total number fish used in the different studies from where the recordings were extracted, as well as the number of individuals/species were recorded and the sex.

Paragraph: Genomic sampling and phylogenetic analyses

Please, clarify whether or not all DNA samples (i.e. downloaded from former study or de novo amplified) used in the present study correspond to the individual identity of specimens used for acoustic samplings.

Paragraph: Sound recordings and bioacoustic analyses

Line 203. Please, indicate how many individuals per species were used if not done previously.

Precise the number of sounds per individuals or at least a range (min and max number of sounds) included in the acoustic analysis.

Line 207-209. It would be judicious to further assess (experimentally or to dig into the literature) that the different equipment’s used to record the different species have only a minimal impact on the acoustic signal of the fish. This is a key sensitive point in bioacoustics that becomes even more critical in the attempt of the present study to sort the acoustic variability according to the phylogeny. The authors may want to seriously consider that aspect as acoustic variability is the half or the core of their work.

Line 209-212. How does the fish emitting the sound was recognised in the resident intruder assay where two individuals are putatively producing sounds, especially in the male-male context?

Line 205-212. Please, provide details and rationales.

Since the recordings were performed during the breeding season how the authors can be sure or justify that the most represented sound picked up from trials including male-female were equivalent behavioural signals across trials, experiments and species, especially in species able to produce two, three or more sounds, i.e. were there agonistic or mating sounds picked-up? This point is crucial since the authors elaborate on sound communication playing “a significant role in mating recognition and prevention of hybridization – line 111” in the introduction and elsewhere in the discussions. In the case where other sound types were recorded it would be interesting to also examine their acoustic structures in the light of the same chosen acoustic parameters and check how they cluster on the PCA and DFA analysis, and further investigate whether they influence the correlation. These additions could interestingly provide evidence to further ground the discussion.

What was the rational for choosing a mix of sounds from both male-female and male-male trials, did all species equivalently underwent those assays? If not, please provide for each species which of the conditions the sounds were extracted from, in a table for instance.

Line 217-219. Can the authors explain why only one type of sound was chosen in species able to produce two or three sound type, despite the criteria of choosing the most represented sound during the trials? Please, see related comment in the discussion section.

Line 223-225. Redundant with line 200-202, please combine or remove.

Paragraph: Comparison between acoustic and genetic data

Line 239. Justify why the outgroup acoustic characteristics was not include in this analysis?

Line 252. “DNA markers”. Same comment as above, line 114.

Paragraph: Statistical analyses

Line 262-264. Please can you explain the necessity to log- and square root-transform data?

Results section

Paragraph: Interspecific acoustic variation and sound properties paragraph

Line 306. Correlation matrix and PCA: Did the authors checked for redundancy to include NP and DUR parameters in the PCA, regarding their coefficient of correlation (p=0.88)? Highly correlated variables may not be included and only one of the variables should be considered to run the PCA analysis; keeping the redundant variable may be accepted when evidence show no significant difference with or without the redundant variables in the PCA score and clustering.

Line 334-337. sDFA: The lowest scores are found in G. cobitis and P. nigricans, species for which only 4 individuals were recorded while G. paganellus, Z. ophiocephalus and P. kessleri show the maximum score of 100% and include 8 to 15 animals. The work would benefit the addition of more specimen for G. cobitis and P. nigricans to minimize the individual effect that would bias the goal of the study focused on acoustic characteristic between species and at least comments on this in the results and discussion. Additionally, the standard deviation of the total length (Raick et al., 2020) should not be neglected especially in small groups (4 animals).

Line 350-352. Please refer to the quadrants according to the XY axis of the DFA to describe the results according to the figure, e.g. P bonelli does not “localise on the right side of the diagram” as stated. Please avoid vague formula like “right side of the diagram” to describe the coordinates/location of the species on the scatter plot instead of. Modify here and elsewhere when appropriate.

Line 358-359. Please rephrase “Zosterisessor ophiocephalus occupies a separate position […]” with appropriate location information.

Line 360. This is unclear why DUR and PRR are considered as the two most important acoustic parameters to generate the figure 3, while NP shows a higher score in the DFA table.

Paragraph: Phylogenetic affinities between vocal gobiids

Line 394. It would be interesting and valuable to integrate non-sonic related species within the genetic distance matrix and tree.

Paragraph: Acoustic and genetic divergence comparison paragraph

Line 430. It is unclear what the cluster analysis refers to and how it was done, please explain here or in material and method section

Line 443-444. The statement “regardless the type of genetic marker” may be modulated, also in the light of a justification for having chosen those four genes.

Discussion section

The overall discussion would benefit to be reorganised for a smoother reading, i.e. avoiding back and forth statements and shortened and some statements should be moderated.

Line 454-457. Learning and memory processes associated with neuroanatomy in teleost fishes where described in former studies and should not be eluded from the present discussion. Please revise the statement. Here again, the physical sound structure may not be confounded with the acoustic behaviour; sounds, either they would be stereotyped or not may not be right away associated with acoustic communication with no direct evidence and the statement “evolution of acoustic communication and its association with phylogeny” although tempting, should be changed.

Line 486-489. Confusing: the authors chose the most represented sound from behavioural trials and did not base their hypothesis on the diversity or number of sound categories within/between species. Thus, conclusions about (equivalent) sound categories with differences in acoustic parameters through species should be draw instead of focusing on the diversity of sound categories within/between species; considering also that some species can only produce one single sound type as mentioned. Rephrase and invert the order of sentences in a way that the particularity already described in P. bonelii for its ability to emit several sound types is now supplemented with the new results of the present study.

Line 491-493. Please consider revising the discussed interpretations in the light of a truthful accurate description of the results: this is not obvious in figure 1, figure 2, neither in figure 3 that Z. ophiocephalus and G. niger share acoustic parameters although they are grouped together in the concatenated phylogenetic tree (e.g. duration, NP, PRR and peak frequency seem different in figure 1). In the PCA and DFA plots as well as in figure 3, Z. ophiocephalus segregates according to the acoustic parameters while G. niger and G. cobitis seem to share 4/6 of the acoustic parameters studied including duration and PRR that are considered here as the most relevant ones. The significant differences between acoustic parameters highlighted by the KW analysis through the studied species should be reported on figure 1 to facilitate the reading, as proposed in the result section for line 311-315.

Line 503-505. Same comment as above for line 486-489. The present study did not focus on the variety of sound type within but between species and only one single sound type was considered even in species able to produce two to three sound types. Please, consider to reorganize the idea developed here.

Line 528. The authors made nice efforts to produce a decent data set of sounds however suggestions mentioned in the materiel and method section must be taken into account prior to claim about its robustness and the statement “robust sound data set to investigate acoustic diversity” could be reasonably moderated.

Line 542-544. “that findings suggest that stochastic forces could be more responsible for shaping acoustic divergence”. The statement here should be tempered regarding the power of the correlation (r=0.47) between acoustic divergence and genetic distance. Also, considering that no alternate analysis showing that acoustic divergence does not follow a non-stochastic hypothesis was tested in the present study.

Line 542-585. “In the evolutionary sense […] since sympatric taxa compete for acoustic niches.” This paragraph should be reorganized more straightforwardly: speculations about adaptation and stochastic processes regarding acoustic behaviours which are enounced with high statements may be modulated, suggestions in the material and methods, and in the results, may be used to modulate the aims exposed in the discussion, statements related to physical structure of acoustic signals and to acoustic communication per se must be wisely clarify especially when associated with evolutionary processes, e.g. genetic-mutation orders and drift or ecological and sexual selection.

Line 586. Replace “built” by “propose”

Line 579-581. “If sexual selection was the main driver of acoustic divergence in gobiids, we would not expect a positive association between song similarity and genetic divergence, since acoustic signals would then diverge faster than genetic loci.” Somehow confusing since the sounds used in the present study were recorded in male-female trials during the breeding period.

Conclusion

Line 609-611. Note. Same comment as above line 542-544. Considering that no alternate analysis showing that acoustic divergence does not follow a non-stochastic hypothesis was tested in the present study.

Figure and Table Legends

Figure1. Report significant differences highlighted by the KW analysis and/or provide associated exhaustive statistical results in a table for instance.

Figure 3. Annotate the units in all representative sonograms.

Figure 4. On the node annotated as -/80, what does the dash mean?

Figure S2. Please describe in the legend what are the plotted dot.

Line 417-418. Unclear why two wave sounds are presented for P. glenii, please clarify.

Table 1. Please precise what are open boxes.

Table 4. Please, clarify the legend or re-annotate the table for a better understanding of the percentages and the meaning of the numbers in columns headed by species.

TableS1. Please provide a better detailed legend describing the table.

Figure S1. Please complete the legend.

Reference section

Please check and modify for the appropriate typo, dates and formats of the provided bibliography throughout the reference section.

Line 656. Typo in Ref 4: replace Malati by Malawi

Line 734. Format Ref 34 with the date of publication

Line 749. Format Ref 41 for the date of publication

Line 757. Ref 44…

6. PLOS authors have the option to publish the peer review history of their article (what does this mean?). If published, this will include your full peer review and any attached files.

Reviewer #1: No

Reviewer #2: No

---

## [Author Response · Author response to Decision Letter 0]

1 Aug 2021

Response to reviewers

Reviewer #1: The authors obtained solid data to support their main conclusions. The overall writing of this manuscript is good except for few typos or errors (such as Lines 23 & 42: "Teleost" should be "teleost"; Lines 220: "implies" should be "imply").

• This was corrected throughout the Manuscript.

Reviewer #2: The present manuscript from Horvatić et al., investigates the relationship between the acoustic variability and the phylogenetic distance of sonic teleost fish, European gobiids. The authors analysed the spectro-temporal acoustic properties of nine vocal gobiids from pre-existing audio materials recorded in other studies to assess their variability. They constructed a phylogenetic relationship between these nine sonic gobiids based on concatenated genetic sequences from mitochondrial and nuclear genes. Finally, the authors confronted the genetic distance with the acoustic divergence to assess whether the degree of correlation could explain the hypothesis that interspecies sound variability falls under stochasticity. Overall, the hypothesis developed in this manuscript is interesting and the detailed procedure provided in the method section is appreciated. However, the study would benefit of additional evidences to support the discussion and conclusions. The terminology of bioacoustics and evolutionary processes must be appropriately defined and used to avoid misinterpretation of the concepts developed here. Some reorganisations in the discussion, corrections and precisions throughout the manuscript, as well as moderation of statements are suggested below. The manuscript would benefit from proofreading for consistency.

• This study aimed to encompass all the acoustic data available from the lineage Gobius, and surely, this material can be considered necessarily heterogeneous in terms of number, methodology and types of sounds collected. However, the main goal was the discrimination (on acoustic and genetic basis) of the species within the lineage, and this discrimination is evident in the results. Despite the variability observed within species, this manuscript discriminates the taxa to the species level. Since the focus of the paper is a phylogenetic analysis, we wanted to capture the main acoustic and genetic features of the species, and it seems that the variation within species and between studies did not mask this species-specific differences.

ABSTRACT

Line 20. Is identity related to individual or species acoustic identity? Please precise.

• Here we included species identity, since some studies have emphasised that certain fish species, such as mormyrids, sciaenids, gobies, etc., can be recognized according to their sounds (Crawford et al., 1997; Amorim et al., 2004; Parmentier et al., 2009, 2018; Bolgan et al., 2020).

Line 30. Acoustic communication should be replaced by acoustic signals. No behavioural experiment was conducted in this study; see related comment below in the Method section - sound recordings and analyses.

We agree and this was corrected in the Manuscript. We carefully replaced the term communication with other more appropriate terms (sound emission, production, etc.).

Line 36. Please, rephrase to better discriminate between what was "invoked" by others studies versus what is corroborated in the present study.

• Corrected in the Manuscript. Since the complete abstract was slightly revised, this sentence was deleted. 

INTRODUCTION SECTION

Line 56, 57, 62. Why using alternatively double and single inverted coma? Please homogenise throughout the manuscript if no specific significance.

• Corrected. We used the same type of inverted comma (“) throughout the Manuscript. 

Line 68. Typo: support instead of supporting.

• Corrected in the Manuscript.

Line 69. If "patterns" refer to acoustic variation and genetic divergence please replace by a more global and inclusive word, e.g. notions, components...

• Corrected in the Manuscript, we replaced it with divergences as this is a more appropriate term. 

Line 71-76. Please consider rearranging/rephrasing: The vocal repertoire among Teleost fishes has been thoroughly investigated. In the gobioids (Gobiiformes; Gobioidei), soniferous species produce different types of acoustic signals consisting of a rich repertoire composed of pulsatile and tonal sounds. This acoustic repertoire shows a great variability at both the inter- and intraspecific levels, with four different sound types [thump, pulsatile (drum), tonal, and complex] recorded to date, emitted mainly by males as part of the breeding and aggressive sonic behaviours.

• Corrected in the Manuscript as suggested. 

Line 73-76: Are females not able to produce sounds in any of the cited behavioural contexts, or in general?

• In some gobies, such as monkey goby (Neogobius fluviatilis), females have the ability to produce sounds during aggressive (female-female) interactions. This was corrected in the Manuscript and an explanation included. 

Line 76. British English: behavioural; please homogenise throughout the manuscript.

• Corrected in the Manuscript. We corrected and included terms behaviour/behavioural. 

Line 77. Redundant, choose either one: ...like in, for example,...

• Corrected in the Manuscript. 

Line 80-81: American English: specialization, utilize; please homogenise throughout the manuscript.

• This was corrected in the Manuscript. Utilization/specialization was replaced by utilisation/specialisation. 

Line 80. Please quickly explain what "lack of sonic specializations" refers to.

• Agree, this is very useful to include in the Manuscript. Corrected in the Manuscript by including the sentence with examples (such as ridged pectoral spines, pharyngeal teeth, enhanced pectoral fin tendons or sonic muscles attached to the swim bladder). 

Line 81. "currently", confusing and may be removed.

• Removed.

Line 82. Note: Vocal communication commonly refers to sounds produced with respiratory system (the larynx in mammals and reptiles or the syrinx in birds). Some acoustic studies in fish use that term to define precise sound types often associated with mechanisms involving air pathways as the swim bladder. Thus, vocal communication, vocalisation and call terms should be wisely used or must be replaced by more general terms, e.g. sonic or acoustic communication, as the present manuscript refers to various sound types from species sometimes lacking the swim bladder.

• Corrected. Vocal, call or vocalizations was replaced throughout the Manuscript with appropriate terms (soniferous, sounds, acoustic signal, etc.). 

Line 83-100. Please reorganise this paragraph to facilitate the reading. Replace or precise "species-rich groups" with the appropriate and consistent terminology, e.g. when putting into opposition "[...] Gobius-lineage form a monophyletic group [...] whereas the genus Gobius [...] the most species-rich groups."

• We agree, the most species-rich group has been changed into most speciose group. The sentence regarding the taxonomic groups has been rephrased. 

Line 93-100. The description of the percentage and proportion of sonic species in each group and lineages is difficult to follow. Similarly, the authors are describing and citing published studies in a way that one could understand they are describing experiments performed in the present study.

• We agree, this paragraph has been deleted. 

Line 93. Are experimental trials referring to behavioural trial? If so, please quickly precise or replace experimental by behavioural.

• Corrected (experimental was replaced into behavioural). 

Line 96. "vocally", please see comment above, line 82.

• We agree, the term vocal was replaced with soniferous, acoustic or sonic when appropriate. 

Line 97. Are bioacoustics experiments trials referring to behavioural trial? Same comment.

• Corrected (bioacoustics was changed into behavioural). 

Note. Please make sure to discriminate between the capability of fish to produce sounds and acoustic communication per se and to use each concept in a more accurate manner.

• We agree, communication has been replaced with the appropriate term (capability, emission, production, etc.). 

Line 103. to a certain degree, please modify.

• Revised. 

Line 102. "call" please see comment above, line 82.

• Corrected (term call was replaced with terms sound or acoustic signal). 

Line 105. "other fish groups". A little vague, replace or precise "groups" with the appropriate and consistent terminology.

• We agree, we have included the part “such as, Malawi cichlids (Cichlidae), toadfishes (Batrachoididae), Dascyllus damselfish (Pomacentridae) or piranhas (Serrasalmidae)”, as it refers to the cited literature.

Line 111. American English: hybridization.

• Corrected to hybridisation. 

Line 111. "vocal" please see comment above, line 82.

• Corrected.

Line 112. American English: specialization.

• Corrected.

Line 114. "genetic markers". Precise to what the term marker refers to here or replace, or remove.

• The term genetic marker is usually used when one is referring to the gene or a part of a DNA which was sequenced. Here we replaced genetic marker with molecular marker (i.e. mitochondrial or nuclear gene fragments), which is the standard terminology in goby literature (Thacker & Roje, 2011; Agorreta et al., 2013). Together with the term molecular marker we also included a precise explanation of its meaning (i.e. DNA fragments/gene sequences). 

Line 117. "in vocal Gobius lineage gobies". Redundancy, gobies should probably be removed.

• Corrected.

Line 120. "multiple genetic markers". Same comment as above, line 114. The authors also may want to elaborate in the introduction which genetic markers they chose and the rational for choosing two mitochondrial and two nuclear genes, how and why they were chosen?

• This part was elaborated later in the M&M (Genomic sampling and phylogenetic analysis), and the intention was to give a short overview of the method(s) in the Introduction (Aims), without listing all the genes separately. Later, in the M&M, where we explained which genes we used in more details (“Samples were amplified for mitochondrial genes cytochrome b (cytb) and cytochrome c oxidase subunit I (cox1), and for nuclear genes Recombination activating gene 1 (rag1) and Rhodopsin (rho)”), we included the sentence in order to follow your comment: “These four genes were chosen here due to their wide application in phylogenetic goby studies [42,45,46,62]”.

Line 129-131. Note. Points iv and v may be moderated and perhaps rethought and rephrased in the light of the comments in the Discussion section (see Discussion section).

• After reading the comments, we still believe that our goals were to build the hypothesis and to discuss potential forces affecting the observed acoustic divergence. Therefore, we would like to keep this part without significantly changing the aims of the current study. However, we have corrected the aim iv. in order to facilitate reading in line with the comments made in the discussion. 

Line 131. "acoustic communication". Please see comment above, line 82.

• Corrected.

MATERIAL AND METHOD SECTION

Paragraph: Study species (Replace by Studied species)

Line 135-140: Please, remove and reorganise with less redundancy with other paragraphs of this section and for a clearer understanding of the rational for choosing the nine species as suggested hereafter: This study analysed acoustic signals and our species composition was based on the availability of audio tracks enabling the comparison of nine soniferous gobiids (Gobiidae, Gobius lineage) belonging to five genera (Gobius, Padogobius, Zosterisessor, Neogobius & Ponticola). Amur sleeper [...]

• Corrected as suggested. 

Line137-139. "The sounds were previously recorded and described by the authors of the present study (see 'Sound recording and bioacoustic analyses') ..." This statement could be introduced in the acoustic recording and bioanalysis paragraph, with the reference of the original study where the audio material was recorded from and/or already used.

• Corrected as suggested, we included the sentence under the proposed subtitle (“Sound recordings and bioacoustic analyses”). Since the sentence continues directly to the other sentence containing the cited references (For all investigated taxa, audio recordings were obtained from laboratory studies [34,49,50,48,80,81,82],), we believe it is not necessary to point the references in the first sentence. 

Line 151. Does "per. Obs." Means personal observations? This paragraph would benefit the addition of the total number fish used in the different studies from where the recordings were extracted, as well as the number of individuals/species were recorded and the sex.

• The per. obs. stands for personal observation, since this was really our field observation. Furthermore, regarding the second part of this comment, information about the number of fish and individuals is located in Supporting file, Table S1, where we have indicated the N (number of the recorded individuals) and n (number of analysed sounds) for ten gobioids. We have included the sentence in the Manuscript which refers to the Table S1 regarding the total number of recorded individuals per species (N). For the rest, all relevant information can be found under the subtitle “Sound recordings and bioacoustics analyses” in the sentence: “Our acoustic dataset consists of 67 soniferous Gobius lineage individuals (for nine species) for which at least ten sounds were recorded per individual and the individual means for each variable were calculated (mean ± SD = 87.0 ± 33.7 sounds analysed per species). Briefly, sounds were recorded from males….”.

Paragraph: Genomic sampling and phylogenetic analyses

Please, clarify whether or not all DNA samples (i.e. downloaded from former study or de novo amplified) used in the present study correspond to the individual identity of specimens used for acoustic samplings.

• We agree and have added the explanation under the proposed subtitle (“For some species from the present study, genetic samples (cytb sequences) do not correspond directly to the soniferous individual (i.e., they do not belong to the fish used for sound analysis), while for some species this was the case (Table 1). For other genes (cox1, rag1 and rho), sequences were designed by the authors or were taken from GenBank®, and they do not correspond directly to the soniferous individual.”). In addition, we have reorganized Table 1 with certain explanations (i.e. asterisks relating the genetic and acoustic samples) to facilitate reading. Since some behavioural (acoustic) experiments were conducted in the past (without collecting the genetic material, such as fin clips, for the subsequent analyses), it was impossible to collect the direct genetic material (i.e. from the soniferous fish), so we tried to follow the protocol in which the genetic material for the current study should come from the exact or closest location compared to the previously (acoustically) tested population/species, if possible. For three gobies, P. nigricans, G. cobitis and G. paganellus, new genetic sequences (cytb gene) come from the same water drainage (i.e. west Italian coast or North Adriatic), but not from the precise location where individuals for acoustic experiments were collected (Serchio River/Venice lagoon). For other species, sequences can be directly linked to the soniferous individual (P. glenii, N. fluviatilis, N. melanostomus) since they were taken immediately after the completion of the acoustic experiments. For P. bonelli, sequences come from completely different water drainage (Zrmanja River). For Z. ophiocephalus, G. niger and P. kessleri, due to technical problem with keeping the stored genetic material for subsequent genetic analysis, DNA sequences (cytb, cox1, rag1, rho) were downloaded directly from genetic sequence database (GenBank®). However, in the case of genetic material used in this study, each DNA sequence correspond to the “taxonomically accurate” goby species, so our phylogram based on the fresh or downloaded sequences reflect the true relationships between the investigated species. We also used phylogenetically stable markers (genes) that reflect the precise relationships between investigated gobies. Lastly, since we did not used a large-scale genetic dataset for investigating the population genetics of soniferous species (many sequences of different individuals), we believe our phylogeny highlights the true affinities between the investigated taxa. 

Paragraph: Sound recordings and bioacoustic analyses

Line 203. Please, indicate how many individuals per species were used if not done previously. Precise the number of sounds per individuals or at least a range (min and max number of sounds) included in the acoustic analysis.

• As stated above, we believe that this information should be summarized in the Supporting file (Table S1), and for that reason, the Manuscript includes only crucial information, while the other details can be found in Table S1. Please see Table S1.

Line 207-209. It would be judicious to further assess (experimentally or to dig into the literature) that the different equipment's used to record the different species have only a minimal impact on the acoustic signal of the fish. This is a key sensitive point in bioacoustics that becomes even more critical in the attempt of the present study to sort the acoustic variability according to the phylogeny. The authors may want to seriously consider that aspect as acoustic variability is the half or the core of their work.

• We agree, and we will explain the procedure below. It is undeniable that different experimental methods and various recording setups can have a significant impact on the experiment outcomes and on our understanding of sound variability or species acoustic differences. However, since the sound recording protocols from the current bioacoustics experiments were set up and practiced by the authors of the present study (S. Malavasi and S. Horvatić), we believe that the experimental conditions in which each fish was recorded are quite similar. We followed several rules in order to fulfil this assumption: 1. Each fish was recorded at a distance of ±15 cm from the hydrophone. The hydrophone was situated in the middle of a large tank (90, 120 or 360 L), placed directly above the shelter occupied by the soniferous male. It is known that sounds can be distorted or attenuate over longer distances, so by minimizing the distance between fish and the hydrophone, we optimized the signal-to-noise ratio (i.e. many sounds from our study were of good “acoustic quality” and could be further manually processed for the analysis). In addition, the recording tank, where we acoustically tested the soniferous fish, was always situated on noise absorbing material (several layers of foam rubber or Styrofoam) to reduce resonance and reflection. Therefore, we minimized ground and air noise, which could disturb the signal recording protocol. 2. The sound system always consisted of two main recording components: hydrophone (preamplified or with external amplifier) and a digital hand recorder. Therefore, the quality of a digital signal was substantially “conserved” throughout the recording experiments. 3. The sound processing protocol followed the same principles in order to produce comparable acoustic signals available for the acoustic analysis (i.e., we used sound amplification for +20 dB, digital formatting of the signal from 44160 Hz to 4000 Hz and 16 bit, and noise filter in the range 50-1500 Hz). Following every step from the previous text, we believe that our representative acoustic data set could indeed reflect the pattern of the acoustic diversity among gobiids. 

Line 209-212. How does the fish emitting the sound was recognised in the resident intruder assay where two individuals are putatively producing sounds, especially in the male-male context?

• Corrected. We have included the sentence: “The soniferous individual was recognized during the experiments according to the presented behaviour, colour pattern or other body characteristics (fin shape or mouth colour).”.

Line 205-212. Please, provide details and rationales. Since the recordings were performed during the breeding season how the authors can be sure or justify that the most represented sound picked up from trials including male-female were equivalent behavioural signals across trials, experiments and species, especially in species able to produce two, three or more sounds, i.e. were there agonistic or mating sounds picked-up? This point is crucial since the authors elaborate on sound communication playing "a significant role in mating recognition and prevention of hybridization – line 111" in the introduction and elsewhere in the discussions. In the case where other sound types were recorded it would be interesting to also examine their acoustic structures in the light of the same chosen acoustic parameters and check how they cluster on the PCA and DFA analysis, and further investigate whether they influence the correlation. These additions could interestingly provide evidence to further ground the discussion. What was the rational for choosing a mix of sounds from both male-female and male-male trials, did all species equivalently underwent those assays? If not, please provide for each species which of the conditions the sounds were extracted from, in a table for instance.

• We agree with the comment, and this will be explained in more detail. For some species, such as G. paganellus and P. bonelli, we did not possess the complete acoustic dataset (pulsatile for G. paganellus and tonal and pulsatile sounds for P. bonelli). Therefore, we selected the representative sounds type and constructed acoustic divergence from the available data. We followed the principle if the species produces one sound type most of the time (during inter- and intrasexual interactions), this type should be considered the most parsimonious character, while the other types can be considered derived states. Our results showed that even these (representative) sounds have the ability to follow the pattern of genetic diversification. Likewise, we believe that our acoustic analysis should be accepted in the present format, with several corrections as proposed. The males are the sound emitting gender in the present study, as has been stressed several times. 

Line 217-219. Can the authors explain why only one type of sound was chosen in species able to produce two or three sound type, despite the criteria of choosing the most represented sound during the trials? Please, see related comment in the discussion section.

• As stated above, we selected the most representative sound type from each species since, firstly, we did not possess the complete gobiid acoustic data set, and secondly, to be able to build the acoustic divergence matrix (which is in fact 9x9 matrix constructed from representative sound types which was correlated with the genetic 9x9 matrix). Sooner or later, within the analysis we would have to “exclude” additional sound types from the species able to produce two/three sound types in order to perform the correlation between acoustic (representative sounds) and genetic (gene sequences) data set. In addition, we believe that some sound types are not as phylogenetically important as others (as previously proven in some animal species), i.e., those that are usually produced during aggressive interaction are considered as deterrent behavioural tool, helping the soniferous fish to defend the territory or repel intruders. 

Line 223-225. Redundant with line 200-202, please combine or remove.

• Corrected. 

Paragraph: Comparison between acoustic and genetic data

Line 239. Justify why the outgroup acoustic characteristics was not include in this analysis?

• The outgroup from our study (Odontobutidae, Perccottus glenii) was intentionally excluded from the comparison analysis for several reasons. First, it was used as an outgroup in the phylogenetic analysis dealing with the construction of phylogenetic trees. In this case, the evolutionary more distant (or ancestral) group, such as Odontobutidae, rooted our concatenated phylogenetic tree. Second, the aim of the present study was the mutual correlation between acoustic divergence and genetic distance in nine sonifeours Gobius lineage gobiids, and not in the whole gobioid group since we do not have insight or the dataset of other additional groups such as Butidae, Eleotridae, etc. If we included P. glenii in the acoustic/genetic correlation, the significant evolutionary and phylogenetic gap between Odontobutidae and the rest of the Gobius lineage gobiids would affect the genetic distance matrix and the interspecific acoustic/genetic comparison would be substantially destructed. Therefore, we decided to exclude P. glenii from the comparative analyses since it is a more distant relative to the investigated group (Gobius lineage, Gobiidae family). 

Line 252. "DNA markers". Same comment as above, line 114.

• Corrected accordingly. See the previous comment, line 114. 

Paragraph: Statistical analyses

Line 262-264. Please can you explain the necessity to log- and square root-transform data?

• This part was explained in the same paragraph by the sentence “We transformed the overall acoustic dataset and tested it for the distribution fitting.” To be more specific, when we applied the log10 and square root transformation, we tested the dataset for the normality by using Shapiro-Wilks W test. Since, after the transformation, we did not achieve the normality of the data, for the interspecific comparison we used the non-parametric Kruskal-Wallis H test. That was the main purpose of the data transformation. 

RESULTS SECTION

Paragraph: Interspecific acoustic variation and sound properties paragraph

Line 306. Correlation matrix and PCA: Did the authors checked for redundancy to include NP and DUR parameters in the PCA, regarding their coefficient of correlation (p=0.88)? Highly correlated variables may not be included and only one of the variables should be considered to run the PCA analysis; keeping the redundant variable may be accepted when evidence show no significant difference with or without the redundant variables in the PCA score and clustering.

• We agree, therefore, we have excluded the redundant variable (number of pulses; NP) from the PCA analysis. Additionally, to check for the differences, we performed two separate PCA analyses: one with and other without NP. The taxa clustering in the scatterplot did not change significantly (species composition was more or less the same) between the two analyses, though the factor loadings for these axes (i.e. factor coordinates) changed significantly. Specifically, in the analysis with NP, the PC1 (35%) was loaded with NP (0.9), DUR (0.7) and PRR (0.7), while the PC2 (27%) with FM (-0.7) and PF (0.6). In the second analysis (without NP), factor loading changed, with PC1 (40%) being loaded with FM (0.8) and PRR (0.7), and PC2 (22%) with DUR (-0.9) and PRR (-0.3). Therefore, we included the results from the analysis which excluded the variable NP. Since fish size (TL) could have a significant impact on the acoustic variables, we performed additional PCA analysis including five XTL-1 variables. 

Line 334-337. sDFA: The lowest scores are found in G. cobitis and P. nigricans, species for which only 4 individuals were recorded while G. paganellus, Z. ophiocephalus and P. kessleri show the maximum score of 100% and include 8 to 15 animals. The work would benefit the addition of more specimen for G. cobitis and P. nigricans to minimize the individual effect that would bias the goal of the study focused on acoustic characteristic between species and at least comments on this in the results and discussion. Additionally, the standard deviation of the total length (Raick et al., 2020) should not be neglected especially in small groups (4 animals).

• Regarding the number of individuals used in this study, the present data set offers the most comprehensive acoustic taxa composition available at this time. Since additional sounds, in the meantime, were not recorded from additional individuals of the soniferous gobiids including the two mentioned taxa (such as G. cobitis or P. nigricans), and considering the fact that behavioural (bioacoustic) experiments are usually performed in a highly repetitive manner and during the prolonged time period (they can last for couple of months until appropriate, high-quality sounds are collected), additional samples could not be included in the present study. This smaller sample size of G. cobitis and P. nigricans compared to P. kessleri or G. paganellus will be noted in the Results and discussed in Discussion. Regarding the second part of this comment, we agree with the suggestion that some acoustic features could be affected by the body size of the emitter. Therefore, we carried out new statistical analyses taking into account the TL of our fish, following the formula “XSL-1” (X is an acoustic variable) presented by Raick et al.(2020). In our case, the difference compared to the Raick et al. (2020) formula is that we used TL instead SL, since for the majority of the species from the present study we do not have the SL measures. For these purposes, we performed two separate comparisons, with original acoustic features and with XSL-1 feature. Kruskal-Wallis test proved that all XSL-1 features differed significantly between each species, as already reported when using non-transformed data for interspecific comparison as presented before. Therefore, in the Manuscript we will report that data do not differ between uncorrected TL method and XSL-1 method (“This pattern was also observed even after we removed the effect of the fish size (TL in mm) on the acoustic variables by dividing them with TL (Kruskall-Wallis test, P < 0.001; d.f. = 9, n = 73 for each sound property”), but will keep the graphical data (Fig. 1. and Fig. S1) from the initial Manuscript. In addition, the PCA analysis was also performed with and without TL correction and the results were similar with slight differences in factor loadings (PC1 loaded with -0.85 PRR and –0.82 DUR; PC2 with -0.93 FM and 0.48 PF) and cumulative percentage for first and second axis (53.3% and 26.2%, respectively).

Line 350-352. Please refer to the quadrants according to the XY axis of the DFA to describe the results according to the figure, e.g. P bonelli does not "localise on the right side of the diagram" as stated. Please avoid vague formula like "right side of the diagram" to describe the coordinates/location of the species on the scatter plot instead of. Modify here and elsewhere when appropriate.

• Agree, corrected accordingly throughout the Manuscript. We refer to parts of DF1 and DF1 axes or coordinates of DF1/DF2. 

Line 358-359. Please rephrase "Zosterisessor ophiocephalus occupies a separate position [...]" with appropriate location information.

• This sentence was removed from the prepared Manuscript.

Line 360. This is unclear why DUR and PRR are considered as the two most important acoustic parameters to generate the figure 3, while NP shows a higher score in the DFA table.

• We have reanalysed the data and performed the statistical analysis once again, and we agree with this comment. Therefore, we have changed this part in Results and Discussion accordingly, where we refer to PRR vs NP scatterplot, plotted with the two most important variables from sDFA. 

Paragraph: Phylogenetic affinities between vocal gobiids

Line 394. It would be interesting and valuable to integrate non-sonic related species within the genetic distance matrix and tree.

• We agree, however, the aim of the present study was to investigate the interspecific acoustic and genetic relationships between soniferous European species, without the violation of the phylogeny with the large diversity of Gobius lineage species (and subspecies) recognized today. By including non-related taxa into phylogenetic (and acoustic) analyses, crucial information (and probably phylogenetic signal) regarding the evolution of the sounds could have been lost or misinterpreted. An additional problem could be the large phylogenetic diversity and taxonomic relationships of European gobies (including Gobius lineage), which are highly complicated and largely (still) unknown even to the experts who study their ecology and phylogeny. Lastly, a comprehensive phylogenetic paper, including some of these species, is currently being preparation by some of the authors of the present study. Therefore, in this study we used only the soniferous species for which we had available data set composed from previously recorded sounds, and the genetic material (gene sequences) produced by us or instantly available from Genbank. 

Paragraph: Acoustic and genetic divergence comparison paragraph

Line 430. It is unclear what the cluster analysis refers to and how it was done, please explain here or in material and method section

• We have corrected this in the Results by including “Cluster analysis (Joining tree analysis) in STATISTICA®”. The other details regarding the analysis can be found in the M&M part. 

Line 443-444. The statement "regardless the type of genetic marker" may be modulated, also in the light of a justification for having chosen those four genes.

• Agree, corrected accordingly. 

DISCUSSION SECTION

The overall discussion would benefit to be reorganised for a smoother reading, i.e. avoiding back and forth statements and shortened and some statements should be moderated.

• We agree and for the purpose of smoother reading, we divided the Discussion into several subtitles and the text into additional paragraphs. We excluded redundant parts of the text and reorganized the Discussion to facilitate reading. In addition, new parts of the Discussion were included. Lastly, by including new analysis dealing with ancestral states for sound types (Mesquite), we gained new insights into the acoustic diversity in gobiids. 

Line 454-457. Learning and memory processes associated with neuroanatomy in teleost fishes where described in former studies and should not be eluded from the present discussion. Please revise the statement. Here again, the physical sound structure may not be confounded with the acoustic behaviour; sounds, either they would be stereotyped or not may not be right away associated with acoustic communication with no direct evidence and the statement "evolution of acoustic communication and its association with phylogeny" although tempting, should be changed.

• The part with the cofounded effect of learning was deleted from Manuscript. We agree that the ability of sound production cannot be mistaken for acoustic communication. Therefore, the communication was also excluded from most part of the Manuscript, considering your previous comment(s). 

Line 486-489. Confusing: the authors chose the most represented sound from behavioural trials and did not base their hypothesis on the diversity or number of sound categories within/between species. Thus, conclusions about (equivalent) sound categories with differences in acoustic parameters through species should be draw instead of focusing on the diversity of sound categories within/between species; considering also that some species can only produce one single sound type as mentioned. Rephrase and invert the order of sentences in a way that the particularity already described in P. bonelii for its ability to emit several sound types is now supplemented with the new results of the present study.

• We sincerely hope that we have understood this comment correctly, therefore, it will be elaborated below. It is clear that some species of gobiids produce a single sound type, while others are able to produce two or even three different types. However, as stated in the Results “However, the nine soniferous gobiids share certain common characteristics of their vocal repertoire, allowing for interspecific comparison”, the interspecific comparison was achieved following strict calculation protocols. The acoustic and genetic correlation implies that the correlation matrices should be structurally the same (9x9, 10x10, etc.) and therefore, we had to choose the acoustic dataset available for the comparison with genetic data. In addition, in our study, a positive correlation between genetic and acoustic divergence was achieved using even representative sound types, which justifies our hypothesis that sounds could reflect phylogenetic relationships. There are several examples (Malavasi et al., 2008; Rice and Bass, 2009) where the authors compare the sounds in certain fish groups (even gobies), even though the sounds are not structurally the same (they do not present the same waveform, have different frequency ranges, etc.). Therefore, although we performed the correlation of acoustic and genetic distance on an “isolated” or “incomplete” dataset (we used the sound variables from most representative sound types), taking into account the known sound diversity in gobies, we believe it is reasonable to discuss different sound types (or as you refer, “categories”) and their evolutionary, ecologically and other significance in the Discussion. We justified our findings by revising the text and including the explanation for acoustic diversity. 

Line 491-493. Please consider revising the discussed interpretations in the light of a truthful accurate description of the results: this is not obvious in figure 1, figure 2, neither in figure 3 that Z. ophiocephalus and G. niger share acoustic parameters although they are grouped together in the concatenated phylogenetic tree (e.g. duration, NP, PRR and peak frequency seem different in figure 1). In the PCA and DFA plots as well as in figure 3, Z. ophiocephalus segregates according to the acoustic parameters while G. niger and G. cobitis seem to share 4/6 of the acoustic parameters studied including duration and PRR that are considered here as the most relevant ones. The significant differences between acoustic parameters highlighted by the KW analysis through the studied species should be reported on figure 1 to facilitate the reading, as proposed in the result section for line 311-315.

• Corrected this part in both the graphical and textual data. We agree that part of the information was misinterpreted, and therefore, instead of three, we recognized five acoustic groups/clusters in the DFA after carrying out additional analyses. The relationships were interpreted accordingly in the Manuscript. 

Line 503-505. Same comment as above for line 486-489. The present study did not focus on the variety of sound type within but between species and only one single sound type was considered even in species able to produce two to three sound types. Please, consider to reorganize the idea developed here.

• Corrected. In addition, the newly included data should enlighten the situation about the sound diversity and acoustic repertoire in different species with the ability to produce various sound types. 

Line 528. The authors made nice efforts to produce a decent data set of sounds however suggestions mentioned in the materiel and method section must be taken into account prior to claim about its robustness and the statement "robust sound data set to investigate acoustic diversity" could be reasonably moderated.

• We agree, therefore, we have corrected this part in the Discussion. The term robust has been excluded, while we continue to use representative when explaining the acoustic diversity. In order to support out theory dealing with different sound types in various gobies, we have included the new section with empirical findings (Mesquite part) in M&M and Results/Discussion. 

Line 542-544. "that findings suggest that stochastic forces could be more responsible for shaping acoustic divergence". The statement here should be tempered regarding the power of the correlation (r=0.47) between acoustic divergence and genetic distance. Also, considering that no alternate analysis showing that acoustic divergence does not follow a non-stochastic hypothesis was tested in the present study.

• We agree, therefore, we have corrected this part in Discussion. Specifically, we have revised the findings regarding the evolutionary forces, especially from the point that the range of correlation coefficient for acoustic and genetic divergence is relatively weak (range 0.45 – 0.47) and that we do not have any additional findings (ecology, morphology, etc.) to support stochastic theory. 

Line 542-585. "In the evolutionary sense [...] since sympatric taxa compete for acoustic niches." This paragraph should be reorganized more straightforwardly: speculations about adaptation and stochastic processes regarding acoustic behaviours which are enounced with high statements may be modulated, suggestions in the material and methods, and in the results, may be used to modulate the aims exposed in the discussion, statements related to physical structure of acoustic signals and to acoustic communication per se must be wisely clarify especially when associated with evolutionary processes, e.g. genetic-mutation orders and drift or ecological and sexual selection.

• We agree, therefore, this paragraph was reorganized and rewritten. Some sentences have been deleted. 

Line 586. Replace "built" by "propose"

• Corrected. 

Line 579-581. "If sexual selection was the main driver of acoustic divergence in gobiids, we would not expect a positive association between song similarity and genetic divergence, since acoustic signals would then diverge faster than genetic loci." Somehow confusing since the sounds used in the present study were recorded in male-female trials during the breeding period.

• Deleted. See the previous comment. 

CONCLUSION

Line 609-611. Note. Same comment as above line 542-544. Considering that no alternate analysis showing that acoustic divergence does not follow a non-stochastic hypothesis was tested in the present study.

• Corrected. Some parts were deleted in order to follow the proposed changes in the Discussion and Conclusion.

FIGURE AND TABLE LEGENDS

Figure1. Report significant differences highlighted by the KW analysis and/or provide associated exhaustive statistical results in a table for instance.

• Corrected. We have included lowercase letters in Fig. 2 in order to emphasize statistical differences. 

Figure 3. Annotate the units in all representative sonograms.

• Corrected in the Manuscript and in Fig. 4. We included the unit (kHz) for the frequency axis. 

Figure 4. On the node annotated as -/80, what does the dash mean?

This omission was corrected, the following was added: Nodes with values ≥0.95 for posterior probability and ≥75% for bootstrap support are considered well supported and are depicted; – indicates posterior probability value <0.95.

Figure S2. Please describe in the legend what are the plotted dot.

• We included the sentence “The plots are the coordinates of the relationship” and “The plots represent the coordinates of the relationship between genetic/acoustic distances” in Fig. 7 and Fig S2., respectively. 

Line 417-418. Unclear why two wave sounds are presented for P. glenii, please clarify.

• We agree, therefore, only the representative sound type (thump) was retained in the figure. 

Table 1. Please precise what are open boxes.

• The Box and Whiskers plots for each acoustic variable were coloured in different colour and the description of each statistical value is included in the figure title “The midline represents the median, x marks the mean, box values indicate the 25th and 75th percentiles, while the whiskers indicate minimum and maximum values of the acoustic properties for each species.”). 

Table 4. Please, clarify the legend or re-annotate the table for a better understanding of the percentages and the meaning of the numbers in columns headed by species.

• Corrected.

TableS1. Please provide a better detailed legend describing the table.

The table was better organised and the caption given more detail, as suggested: Partitioning scheme and best-fit models of evolution for data blocks defined by gene and codon position, assessed by PartitionFinder 2 for subsequent phylogenetic analyses in MrBayes and RAxML.

Figure S1. Please complete the legend.

• Corrected. 

REFERENCE SECTION

Please check and modify for the appropriate typo, dates and formats of the provided bibliography throughout the reference section.

Line 656. Typo in Ref 4: replace Malati by Malawi

• Corrected.

Line 734. Format Ref 34 with the date of publication

• Corrected.

Line 749. Format Ref 41 for the date of publication

• According to the online reference database, the publication date for “Phylogeny of Gobioidei and Placement within Acanthomorpha, with a New Classification and Investigation of Diversification and Character Evolution” is 2009. 

Line 757. Ref 44...

• Same as in the previous comment, the publication date for publication “A new species of Gobius (Perciformes: Gobiidae) from the Mediterranean Sea and the redescription of Gobius bucchichi” is 2016.

---

## [Decision Letter · Decision Letter 1]

15 Oct 2021

PONE-D-21-00246R1Correlation between acoustic divergence and phylogenetic distance in soniferous European gobiids (Gobiidae; Gobius lineage)PLOS ONE

Dear Dr. Zanella,

Thank you for submitting your manuscript to PLOS ONE. After careful consideration, we feel that it has merit but does not fully meet PLOS ONE’s publication criteria as it currently stands. Therefore, we invite you to submit a revised version of the manuscript that addresses the points raised during the review process.

Both reviewers are happy with the new manuscript and it should be acceptable for publication once you respond to the few minor changes suggested by Reviewer 2.

We look forward to receiving your revised manuscript.

Kind regards,

Vivek Nityananda

Academic Editor

PLOS ONE

Journal Requirements:

Reviewers' comments:

Reviewer's Responses to Questions

**Comments to the Author**

1. If the authors have adequately addressed your comments raised in a previous round of review and you feel that this manuscript is now acceptable for publication, you may indicate that here to bypass the “Comments to the Author” section, enter your conflict of interest statement in the “Confidential to Editor” section, and submit your "Accept" recommendation.

Reviewer #1: All comments have been addressed

Reviewer #2: (No Response)

2. Is the manuscript technically sound, and do the data support the conclusions?

Reviewer #1: Yes

Reviewer #2: Yes

3. Has the statistical analysis been performed appropriately and rigorously? 

Reviewer #1: N/A

Reviewer #2: Yes

4. Have the authors made all data underlying the findings in their manuscript fully available?

Reviewer #1: Yes

Reviewer #2: Yes

5. Is the manuscript presented in an intelligible fashion and written in standard English?

Reviewer #1: Yes

Reviewer #2: Yes

6. Review Comments to the Author

Reviewer #1: (No Response)

Reviewer #2: The present version of the manuscript PONE-D-21-00246R1 from Horvatić et al., has been revised and the authors conformed with most comments and suggestions. Editions were properly made where authors agreed with the comments, and explanations were provided when the authors’ choice was to not comply with the suggestions. The overall writing is clearer, allowing an intelligible reading flow. Some of the suggested analyses were done, figures were modified, clarifying interpretations. The authors wisely moderated the assumptions and reshaped a more focused discussion leading now to an adequate philosophy of the general conclusion. However, a few remaining points suggested below would be critical to address and consider.

Material and Methods section

Line 132. Please write slightly more explicitly e.g. “availability of previously recorded audio tracks…”

Line 153. Replace “life-history” trait by “morphological trait”

Line 155. Table S1 and Table S2 should be inverted and appear according to the chronology of the reading. Please modify where it applies in the manuscript and supplementary accordingly.

Results section

Line 349. The PF (Hz) also looks associated with PC1 with a score of -0.71. Please, comment unless the value is under an arbitrary threshold set by the authors which should, in that case, be mentioned somewhere, here or in the material and method section.

Line 353. “PCA accounted cumulatively for 80% of variation” Where is this shown, could the authors add that to Table S3 and precise which PCAs were done with or without TL correction.

Phylogenetic affinities between soniferous gobiids: the morphological parameters i.e. swim bladder and numbers of vertebrae should not be disregard while the authors describe the tree in Fig5, in the manner of the environment is quickly commented.

Line 441-447. Overall, please, rewrite that paragraph a little clearer to facilitate the reading i.e. avoid back and forth, repetitions or cross-descriptions of the tree.

Line 442. A little confusing “Two marine gobies…” Please rephrase to clarify that four of marine gobies are spread into two distinct clades, e.g. one clade with Gobius niger and Zosterisessor ophiocephalus and another one including G. cobitis and G. paganellus.

Line 445-447. Confusing, please rephrase clearer “In the third group, P. kessleri […] in regard to these two species.”

Discussion section

Line 528. Please, add as suggested to remove any ambiguity here as well “…according to their observed frequency in the behavioural experiments the sounds were extracted from.”

Line 550. Change “this” by “our”, or by “the present study” for precision, e.g. “…corroborated by the present study”.

Line 552-553. “This is in agreement with…” The statement does not argue in favour of the findings of the present study showing that P. bonelli ranks separately from the other species described line 550-552. The authors focused on the most represented sound within species, including in P. bonelli that is able to produce other sound types. Since the other sound types were not investigated for that species here, it is not possible to ascertain that acoustic clusters in PCA, DFA or else would have been similar. Please, remove or replace by another argument to discuss that P. bonelli ranks separately.

Figure legends section

Line 367. Fig2. Explain somewhere (in the figure caption for example) what letters correspond to, i.e. significant differences between which species.

Line 414. Fig4. Please annotate in that legend what PRR and NP stand for.

Line 458. Fig5. Explain scale bar 0.2, i.e. one should easily read whether/how the branches are time-scaled.

Line 476. Table 5. Explain the empty box column “1.” for G. cobitis.

Figures

Fig3. Double-check for the accuracy of the numbers of plotted individuals in the DFA, e.g. number of empty triangles showing G. cobitis.

Fig6. Please, increase the size of the legend which is currently difficult to read; similarly, with the name of the species.

Tables

Line 370. Table 2. Could be placed in Suppl data e.g. with Table S3 to facilitate the reading, since the main correlated parameters already appear in the main text. Clarify whether this correlation considers the TL correction.

Suggestion to exchange Table S1 and table S2, the latest is commented earlier in the manuscript.

Table S2. Briefly explain the empty FM box (Hz) for P. glenii somewhere. Double-check that the number of recorded individuals set in Table S2 matches with the number of individuals shown in all analysis, tables and plots and everywhere else in the manuscripts, e.g. G. cobitis.

Table S3. Please check the legend for the accurate number of acoustic variables included in the present PCA and quickly mention the rational for removing NP here. Mention here or somewhere else appropriate when PCAs were done with or without TL correction.

7. PLOS authors have the option to publish the peer review history of their article (what does this mean?). If published, this will include your full peer review and any attached files.

Reviewer #1: No

Reviewer #2: No

---

## [Author Response · Author response to Decision Letter 1]

15 Nov 2021

REBUTTAL LETTER 

• Reviewer #2: The present version of the manuscript PONE-D-21-00246R1 from Horvatić et al., has been revised and the authors conformed with most comments and suggestions. Editions were properly made where authors agreed with the comments, and explanations were provided when the authors’ choice was to not comply with the suggestions. The overall writing is clearer, allowing an intelligible reading flow. Some of the suggested analyses were done, figures were modified, clarifying interpretations. The authors wisely moderated the assumptions and reshaped a more focused discussion leading now to an adequate philosophy of the general conclusion. However, a few remaining points suggested below would be critical to address and consider.

We are satisfied that our resubmitted manuscript, after provided Major revision, follows the suggestions proposed by both reviewer(s). In addition, we are aware that some of the minor errors were still present in this resubmitted version, and in this sense, we gave our best to fulfil the demands of reviewer #2 to strengthen our hypotheses and produce a high quality manuscript within the current revision process. Therefore, we believe that this version of the manuscript will meet the standards and prerequisites for its acceptance in world’s leading scientific journals such as Plos One. 

MATERIAL AND METHODS SECTION

• Line 132. Please write slightly more explicitly e.g. “availability of previously recorded audio tracks…”

We agree, therefore, we have corrected this part as you suggested. 

• Line 153. Replace “life-history” trait by “morphological trait”

We agree, and according to this suggestion, we have corrected this part. 

• Line 155. Table S1 and Table S2 should be inverted and appear according to the chronology of the reading. Please modify where it applies in the manuscript and supplementary accordingly.

We completely agree, and this was corrected accordingly throughout the manuscript. 

RESULTS SECTION

• Line 349. The PF (Hz) also looks associated with PC1 with a score of -0.71. Please, comment unless the value is under an arbitrary threshold set by the authors which should, in that case, be mentioned somewhere, here or in the material and method section.

The value was not under the arbitrary threshold, but was simply unrecognized or neglected by the authors during the initial writing period. Therefore, we have included the PF into the account when discussing the results of PCA analysis. 

• Line 353. “PCA accounted cumulatively for 80% of variation” Where is this shown, could the authors add that to Table S3 and precise which PCAs were done with or without TL correction.

We agree with your comment, and following your suggestions, we have decided to improve Fig. S1 and Table S3 by including the information from two PCAs (one without and one with TL correction). 

• Phylogenetic affinities between soniferous gobiids: the morphological parameters i.e. swim bladder and numbers of vertebrae should not be disregard while the authors describe the tree in Fig5, in the manner of the environment is quickly commented.

We agree, therefore, we have included additional textual explanations within the revised manuscript in order to cover your suggestions regarding morphology (no. of vertebrae and swim bladder) and to facilitate the reading/understanding of Fig. 5. 

• Line 441-447. Overall, please, rewrite that paragraph a little clearer to facilitate the reading i.e. avoid back and forth, repetitions or cross-descriptions of the tree.

We agree, therefore, we have revised this section according to suggestions. 

• Line 442. A little confusing “Two marine gobies…” Please rephrase to clarify that four of marine gobies are spread into two distinct clades, e.g. one clade with Gobius niger and Zosterisessor ophiocephalus and another one including G. cobitis and G. paganellus.

We agree, and we have corrected this part of the manuscript. Now, the sentence formulation goes: “Of the four marine gobies (which share the reduced number of vertebrae [< 28] and the presence of swim bladder), Gobius niger and Zosterisessor ophiocephalus grouped into one clade, while G. cobitis and G. paganellus clustered into the second.”

• Line 445-447. Confusing, please rephrase clearer “In the third group, P. kessleri […] in regard to these two species.”

We agree, therefore, this section was improved according to your suggestions: “The third group was composed of gobiids distributed in the Ponto-Caspian region (genera Neogobius and Ponticola) and P. nigricans, an Italian freshwater endemic species (Fig 5). This group occupies freshwater habitats, has a higher number of vertebrae (> 28) and lacks a swim bladder. Specifically, in the third group, P. kessleri is a sister taxon to the remaining species, while within the “Neogobius group”, Neogobius fluviatilis is a sister taxon in regards to two closely related species, P. nigricans and N. melanostomus”.

DISCUSSION SECTION

• Line 528. Please, add as suggested to remove any ambiguity here as well “…according to their observed frequency in the behavioural experiments the sounds were extracted from.”

We agree, therefore, this part was corrected as suggested.

• Line 550. Change “this” by “our”, or by “the present study” for precision, e.g. “…corroborated by the present study”.

We agree, therefore, this part was corrected as suggested.

• Line 552-553. “This is in agreement with…” The statement does not argue in favour of the findings of the present study showing that P. bonelli ranks separately from the other species described line 550-552. The authors focused on the most represented sound within species, including in P. bonelli that is able to produce other sound types. Since the other sound types were not investigated for that species here, it is not possible to ascertain that acoustic clusters in PCA, DFA or else would have been similar. Please, remove or replace by another argument to discuss that P. bonelli ranks separately.

We agree, therefore, we have eliminated this sentence from the revised manuscript. 

FIGURE LEGENDS SECTION

• Line 367. Fig2. Explain somewhere (in the figure caption for example) what letters correspond to, i.e. significant differences between which species.

If we understand this comment correctly, the proposed suggestion is to explain each letter for each species within Fig 2. If this is the case, we believe that including the explanation for each letter would take too much space in the figure caption, or especially, in the text. In addition, by including the letter explanations, the graphs within the Fig 2. become unnecessary and the figure becomes redundant. From the present figure caption and Fig 2. itself, we believe it should be simple to conclude that species with the letters a, b, c, etc. are significantly different, and that those with a combinations of letters (bc, adc, etc.) share similar properties. To further strengthen our rationale for not including the explanations for each letters in text/caption, we propose two (of many) papers where the similar approach has been followed (Velasquez et al., 2013 “Bioacoustic and genetic divergence in a frog with a wide geographical distribution” & Lee et al., 2016 “Geographic variation in advertisement calls of a Microhylid frog – testing the role of drift and ecology”). Therefore, we strongly believe there is no need for the inclusion of the explanations of the different letters, especially considering Fig 2, which would in that case become redundant. 

• Line 414. Fig4. Please annotate in that legend what PRR and NP stand for.

We agree, therefore, we have included explanation (NP - number of pulses, PRR - pulse repetition rate). 

• Line 458. Fig5. Explain scale bar 0.2, i.e. one should easily read whether/how the branches are time-scaled.

We agree, therefore, we have included the explanation in the figure caption: “Branch length scale represents number of substitutions per site.”

• Line 476. Table 5. Explain the empty box column “1.” for G. cobitis.

After the initial revision, the idea was to highlight the correlation between different species within Table 5, while giving the dash (-) to the states where we observed an intercorrelation within the same species (e.g. G. cobitis vs G. cobitis). Afterwards, we have noticed that the practical solution for genetic distance matrices (as proposed by the MEGA software instructions and as noted in many phylogenetic papers, such as Buj et al., 2017 “Ancient connections among the European rivers and watersheds revealed from the evolutionary history of the genus Telestes (Actinopterygii; Cypriniformes)” or Grzywacz et al., 2017 “Evolution and systematics of Green Bush-crickets (Orthoptera: Tettigoniidae: Tettigonia) in the Western Palaearctic: testing concordance between molecular, acoustic, and morphological data”) is to leave an empty cell with the intercorrelation coefficient within the species (e.g. G. cobitis vs G. cobitis). We have followed the same approach in this revised manuscript, and therefore, we eliminated the dash from Table 5 while simply leaving the cell empty for G. cobitis vs G. cobitis, G. paganellus vs G. paganellus, etc... 

FIGURES

• Fig3. Double-check for the accuracy of the numbers of plotted individuals in the DFA, e.g. number of empty triangles showing G. cobitis.

We have checked and resolved this suggested problem. The “real” number of G. cobitis individuals was always 6, so the initial confusion was produced probably due to Table S1, where we unintentionally indicated that we used 4 individuals of G. cobitis. This was corrected in the revised manuscript (in Table S1 we have corrected 4 into 6 individuals). There are indeed six triangles in the PCA and DFA, meaning that six individuals of G. cobitis were used in our acoustic analyses. 

• Fig6. Please, increase the size of the legend which is currently difficult to read; similarly, with the name of the species.

We agree, and therefore, we have corrected this part as suggested. 

TABLES

• Line 370. Table 2. Could be placed in Suppl data e.g. with Table S3 to facilitate the reading, since the main correlated parameters already appear in the main text. Clarify whether this correlation considers the TL correction.

We agree, therefore, we have moved Table 2 into the Supplementary data, so in the revised manuscript, this table is now referred to as Table S3.

• Suggestion to exchange Table S1 and table S2, the latest is commented earlier in the manuscript.

This was corrected earlier. 

• Table S2. Briefly explain the empty FM box (Hz) for P. glenii somewhere. Double-check that the number of recorded individuals set in Table S2 matches with the number of individuals shown in all analysis, tables and plots and everywhere else in the manuscripts, e.g. G. cobitis.

This was corrected earlier. 

• Table S3. Please check the legend for the accurate number of acoustic variables included in the present PCA and quickly mention the rational for removing NP here. Mention here or somewhere else appropriate when PCAs were done with or without TL correction.

We agree, therefore, we have included the rationale for removing NP within the Table S4 (“For PCAs, we excluded acoustic variable number of pulses (NP) due to its correlation with other variable (DUR)”), and corrected six into five variables. The TL corrections was explained and corrected earlier.

---

## [Editor Report · Decision Letter 2]

18 Nov 2021

Correlation between acoustic divergence and phylogenetic distance in soniferous European gobiids (Gobiidae; Gobius lineage)

PONE-D-21-00246R2

Dear Dr. Zanella,

We’re pleased to inform you that your manuscript has been judged scientifically suitable for publication and will be formally accepted for publication once it meets all outstanding technical requirements.

Kind regards,

Vivek Nityananda

Academic Editor

PLOS ONE
---

## [Editor Report · Acceptance letter]

2 Dec 2021

PONE-D-21-00246R2 

Correlation between acoustic divergence and phylogenetic distance in soniferous European gobiids (Gobiidae; *Gobius* lineage) 

Dear Dr. Zanella:

I'm pleased to inform you that your manuscript has been deemed suitable for publication in PLOS ONE. Congratulations! Your manuscript is now with our production department. 

Kind regards, 

on behalf of

Dr. Vivek Nityananda 

Academic Editor

PLOS ONE